# BayesR3 enables fast MCMC blocked processing for largescale multi-trait genomic prediction and QTN mapping analysis

Edmond J. Breen [1✉], Iona M. MacLeod[1], Phuong N. Ho[1], Mekonnen Haile-Mariam[1], Jennie E. Pryce[1,2], Carl D. Thomas[1], Hans D. Daetwyler[1,2] & Michael E. Goddard[1,3]

Bayesian methods, such as BayesR, for predicting the genetic value or risk of individuals from their genotypes, such as Single Nucleotide Polymorphisms (SNP), are often implemented using a Markov Chain Monte Carlo (MCMC) process. However, the generation of Markov chains is computationally slow. We introduce a form of blocked Gibbs sampling for estimating SNP effects from Markov chains that greatly reduces computational time by sampling each SNP effect iteratively $n$-times from conditional block posteriors. Subsequent iteration over all blocks $m$-times produces chains of length $m \times n$. We use this strategy to solve large-scale genomic prediction and fine-mapping problems using the Bayesian MCMC mixed-effects genetic model, BayesR3. We validate the method using simulated data, followed by analysis of empirical dairy cattle data using high dimension milk mid infra-red spectra data as an example of "omics" data and show its use to increase the precision of mapping variants affecting milk, fat, and protein yields relative to a univariate analysis of milk, fat, and protein.

[1] Agriculture Victoria, AgriBio, Centre for AgriBioscience, Bundoora, VIC 3083, Australia. [2] School of Applied Systems Biology, La Trobe University, Bundoora, VIC 3083, Australia. [3] Faculty of Veterinary and Agricultural Sciences, University of Melbourne, Parkville, VIC 3052, Australia. ✉email: ed.breen@agriculture.vic.gov.au

Many important traits in medicine, agriculture, and evolution are complex and are quantitative traits controlled by many genes and environmental factors[1–5]. Despite the availability of assays for many thousands of Single Nucleotide Polymorphisms (SNPs) for around 20 years, knowledge of the polymorphisms that explain the genetic variation in complex traits is still limited. There are two difficulties in mapping and identifying these causal variants—the effects are typically very small and linkage disequilibria (LD) between polymorphisms make it hard to identify the true causal variants. Consequently, the best strategy is likely to be to estimate the effect of variants simultaneously across the genome, where the effects on the trait, or phenotype, are treated as random variables sampled from a distribution. This analysis is known as genomic selection or genomic prediction (GP)[6,7].

Methods of GP use a mixed-effects linear regression model in which the effect of the marker genotypes, such as SNPs, are treated as random effects, allowing these effects to be estimated even when their number is much greater than the number of phenotypic records. Bayesian methods to do this differ in the prior distribution assumed for the marker effects in this linear model. If they are assumed to be drawn from a normal distribution with mean zero and constant variance, the method is Best Linear Unbiased Prediction (BLUP). However, experimental evidence suggests that many SNPs have no effect, some a small effect and a few a large effect on the trait[8,9]. To represent such a distribution, it is common to use a slab and spike prior and the slab may represent a long-tailed distribution[6,7]. For instance, Erbe et al. introduced a model called BayesR in which the SNP effects follow a mixture of normal distributions including a component with zero variance and effects (i.e., a spike at zero). Genomic prediction methods such as BayesR may provide information about the genetic architecture of complex trait, map the causal variants to regions of the genome, and predict the genetic value of individuals from their genotypes. For instance, this might be a prediction of a person's risk of disease[10–12] or, in agriculture, the genetic value of individual plants or animals so that the best are selected as parents of the next generation[13,14].

SNP effects are commonly estimated using a Markov Chain Monte Carlo (MCMC) algorithm, but MCMC methods are slow. Faster non-MCMC solutions, such as EM algorithms[15–17] or variational Bayes, sacrifice prediction accuracy for speed. Therefore, attention has focused on making the MCMC process more efficient. A recent approach[18] advocated estimating SNP effects in parallel, but it required many computer nodes, hundreds of cores, and twice the expected memory (RAM), thereby restricting its utility to all but the largest computer facilities. Residual updating within a Gibbs sampling scheme is a process analogous to a Gauss-Seidel solution of linear equations in which one SNP is processed at a time, conditional on knowing the other effects, and after the solution is updated, the residuals for all records are updated[19–21]. The updating of all the residuals for every SNP in every Gibbs cycle (iteration) is probably the main reason for the long compute time taken. Calus[19] sped up the Bayesian MCMC procedure SSVS (stochastic search variable selection) by processing 5 or 6 SNPs at a time. Chen et al.[22], exploiting sparsity between markers, produced an updating procedure by keeping $V'V$ in memory, where $V$ is the genotype matrix, and updating SNP effects only when SNP effects were estimated to change. This approach adds significantly to the amount of computer RAM required for analysis, especially as the number of phenotypes and SNPs increase. Still, these MCMC updating procedures remain slower than a BLUP solution[23].

Here, we propose an intermediate strategy in which a block of SNPs is processed together. We present an iterative Gibbs sampling of a block conditional posterior distributions of unknown variables.

We use this approach (called BayesR3) to estimate SNP effects for GP, as in standard BayesR[13]; multi-trait BayesMT[24]; and to perform BayesRC[25] that uses prior information on the SNP effects. We demonstrate that our approach is faster than previous BayesR methods (12) and show its ability to describe the genetic architecture of complex traits, by fine-mapping SNP effects to genomic locations. We apply it to simulated phenotype data, coupled to real genotypes to demonstrate its properties, and then to empirical data on milk yield and composition in a large dairy cattle data set of 75,471 mixed breed Australian bulls and cows with genotypes from an imputed high density 717,463 SNP set. Further, we apply the method to high dimension milk mid-infrared spectral data to illustrate its potential application to "omics" data on traits that are intermediate between genotype and phenotype.

## Results

**The genetic model.** The model assumed for the phenotype or trait values is:

$$y = Xu + Vg + Za + e \qquad (1)$$

where $y$ is an $n_R \times 1$ column vector of phenotype values, $n_R$ is the number of records; $X$ is a $(n_R \times n_F)$ incidence matrix, $u$ is a $n_F \times 1$ vector of fixed effects and $n_F$ is the number of fixed effects; $V$ is a coded genotype $(n_R \times n_M)$ matrix, as constructed in the methods, representing the observed genotypes of each individual across $n_M$ markers (see Methods); $g$ is a vector containing the SNP effects, $Z$ is an identity matrix $(n_R \times n_R)$ and $a$ is a vector of random genetic effects not explained by the SNPs with polygenic variance represented as $\sigma_a^2$; such that $a \sim N(0, A\sigma_a^2)$, and $A$ is the relationship matrix. Note also, that $e \sim N(0, W^{-1}\sigma_e^2)$, where $W$ is a diagonal weight matrix as described in the methods.

**BayesR3.** The mathematics used here for BayesR3 is given in detail in the methods, but briefly the SNP effects are modelled by a mixture of four normal distributions with zero mean and increasing variances as specified by:

$$p(g_j | \pi, \sigma_g^2) = \pi_1 \times N\left(0, 0 \times \sigma_g^2\right) + \pi_2 \times N\left(0, 10^{-4} \times \sigma_g^2\right)$$
$$+ \pi_3 \times N\left(0, 10^{-3} \times \sigma_g^2\right) + \pi_4 \times N\left(0, 10^{-2} \times \sigma_g^2\right).$$
$$(2)$$

Where $\sigma_g^2$ is the additive genetic variance explained by the SNPs cumulatively and is estimated from the data. The choice of 4 distributions, is historical[13], but any number of distributions can be included in the mixture if needed. For example, it has been reported that adding the variance group $10^{-5} \times \sigma_g^2$ can help identify SNPs with very small effects if the dataset is very large[26]. Therefore, the allocations values $\left(0, 10^{-4}, 10^{-3}, 10^{-2}\right)$ seen in Eq. (2) are held constant and used to scale the genetic variance and to help fit long-tailed distributions as discussed in Supplementary Note 1. The 10x scaling between the allocation values allows the distributions generated to be relatively smooth and effects can shuffle from one distribution to the next between MCMC cycles. The mixing proportions $\pi$ are also estimated from the data and are assumed to be drawn from a Dirichlet distribution with parameter $= (1,1,1,1)$, a uniform prior, such that any SNP a priori is equally likely to be assigned to any one of the 4 distributions.

The decrease in processing time reported below is mainly achieved by processing SNPs in blocks. Here the marker effects are divided sequentially into $n_B$ non-overlapping blocks such that:

$$Vg = V_1 g_1 + V_2 g_2 + \dots + V_{n_B} g_{n_B} \qquad (3)$$

The number of blocks, $n_B$, is determined from the block size, $n$, and is defined as the least integer greater or equal to $\frac{n_M}{n}$. All blocks

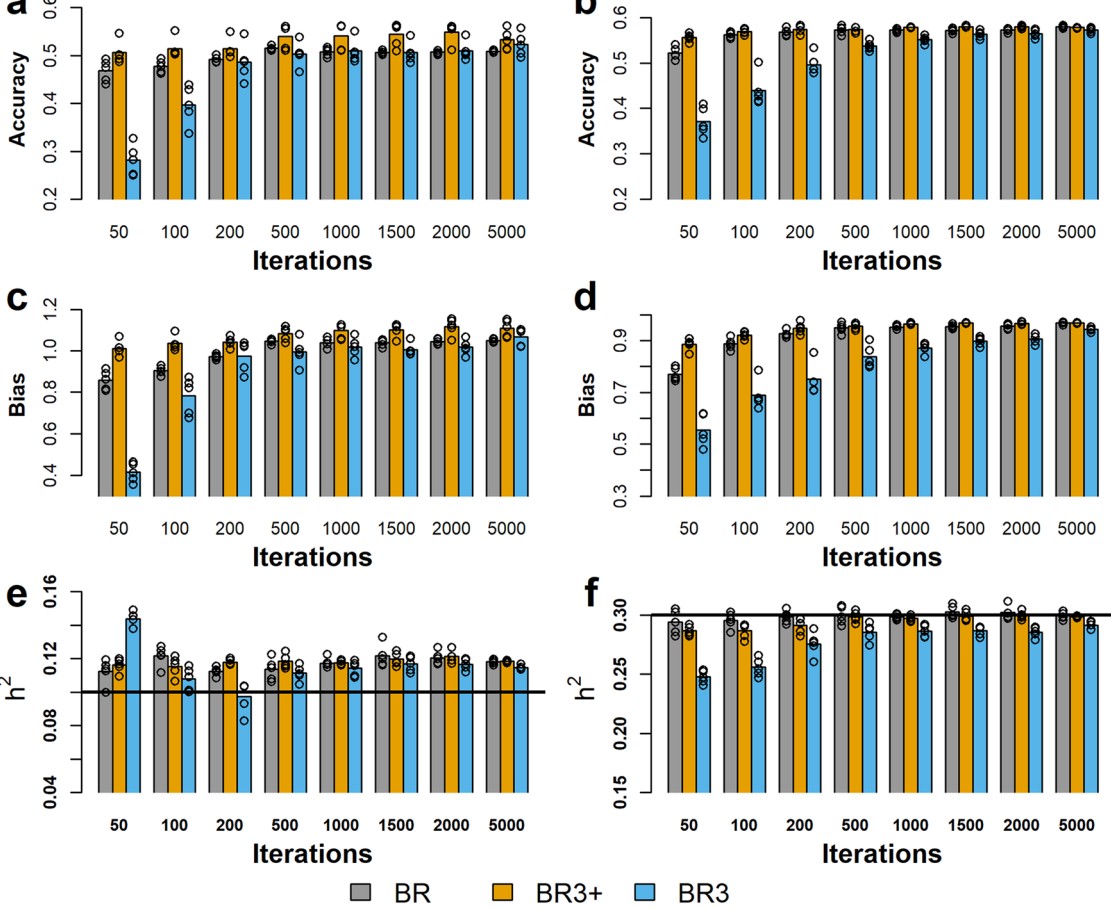

**Fig. 1 Correlation between true breeding value (TBV) and estimated breeding value (EBV) with 10% (H10) and 30% (H30) heritabilities.** X-axis gives the number of iterations performed by BR (grey bars), BR3 (sky blue bars), and BR3+ (orange bars). The bar heights in each plot represents the mean summary statistics obtained from 5 chains and the individual data points from each chain are overlaying on each respective bar. The prediction accuracy for the 2 heritabilities are given in panels **a** for 10% heritability data and in **b** for the 30% heritability data. The prediction biases are given in panels **c** 10% and **d** 30%. Panels **e**, **f** give the estimated 10% and 30% heritability with respect to iteration. The horizontal black line on each of these plots shows the expected heritability for each data set.

are the same size, except the last block that is often smaller. Using Gibbs sampling, the SNP effects in each block are also sampled $n$ times before the next block is processed. Then after $m$ iterations across all blocks, Markov chains of length $n_L = m \times n$ are created for each effect. In other words, the Markov chain consists of $n$ inner cycles within a block and $m$ outer cycles across blocks.

**Analysis of simulated data**. We applied BayesR3 to data consisting of 400,000 SNPs on 20,000 individuals with simulated phenotypes. The SNP genotypes of the 20,000 individuals were real and were taken from 15,220 Holstein and 4780 Jersey cows. We then tested the predicted SNP effects by their ability to predict the simulated true breeding values (TBV) for 1725 Australian Red cows referred to here as RDC using their real genotypes but simulated SNP effects (see Methods). The traits simulated had two different levels of narrow-sense heritability ($h^2$): 0.1 (H10) and 0.3 (H30). The genomic model fitted, also included two fixed effects, one for the mean and the other for breed (Holstein and Jersey). Also, a second set of phenotypes for 41,925 cows from 33,555 Holstein and 8370 Jersey genomes was simulated, which was a superset containing the original 20,000 individuals.

**Prediction accuracy and bias**. We compared BayesR with no blocking and $n_L$ samples drawn per parameter to two versions of

BayesR3. In one version (here called BR3+) block size $n = 25$ SNPs and $m \times n$ samples are drawn per parameter, where $m = n_L$ (i.e., it does the same number of outer iterations as BayesR), and to a version (here called BR3), where $m = \frac{n_L}{n}$, so that the total number of samples drawn is the same as BayesR but only 1/25 of the number drawn from BR3+.

In Fig. 1a, b, the accuracies, as measured by Pearson's correlation between predicted breeding value for each animal and the true simulated breeding value for the 1725 RDC cows are given. The figure compares three BayesR configurations: (1) BR (grey bars) represents a standard "non-blocked" BayesR configuration; (2) BR3 (blue bars) a blocked BayesR, where the number of outer iterations m $= n_L/n$ and $n = 25$; therefore, the Markov Chain lengths are the same as for BR; and (3) BR3+ where: $m = n_L$ and $n = 25$. For BR3+ the results are graphed against the number of iterations divided by 25 so that the accuracy etc of BR3+ and BR can be compared when BR3+ has the same number of outer cycles as BR has total iterations. In summary: BR and BR3 produced chains of the same length, while BR3+ produced chains that were 25 times longer. It is seen in Fig. 1a that BR3+ (shown as orange bars) is consistently more accurate than BR (grey bars), by around 4% across all iteration levels. This is somewhat expected because of the large difference in chain lengths; however, this also proves that the inner cycles are indeed contributing to the result. In contrast, Fig. 1b shows that BR3+ is

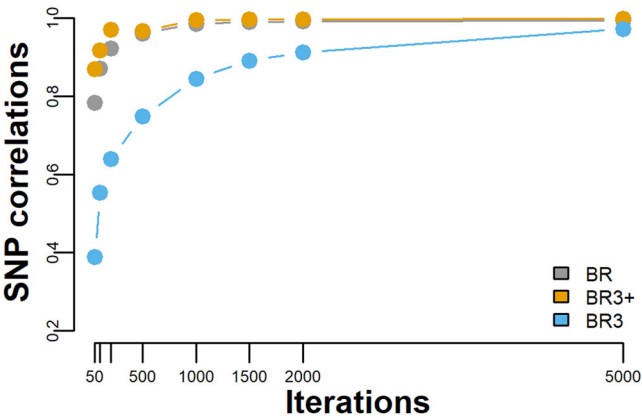

**Fig. 2 Across chain convergence analysis.** Pearson's correlations for iteration {50,100,200,500,1000,2000 and 5000} for the analysis and data presented in Fig. 1b. The number of chains was 5, therefore each plotted point represents the mean of 10 correlations. Results obtained from BR are given in grey, BR3 results are in given in sky blue while the BR3+ results are shown in orange.

only different from BR3 during low iteration counts, after which both chains appear to have reached convergence. Comparing BR (grey bars) with BR3 (blue bars) shows that BR3 requires more total iterations than BR to reach a maximum accuracy but fewer outer cycles because BR3 performs 25 inner cycles per outer cycle. Below we show that it is the number of outer cycles that largely determines the computing time needed.

As well as high accuracy, we would like the estimated breeding values to be well-calibrated. That is, we would like the regression of true breeding value on estimated breeding value to = 1. (This regression is often referred to as "bias" although it is not the classical definition of bias). In Fig. 1c, d these biases are shown, which is given by regression of the TBV on the SNP predicted EBV and where a coefficient of 1 represents no bias. A slope of 1 means the expectation of $u$, given $\hat{u}$ is equal to $\hat{u}$. It is seen in Fig. 1c, d that the bias is generally within 10% of 1, other than at a low number of iterations.

**Convergence.** The estimated SNP effects from an MCMC chain suffer from an error due to sampling. The limited number of samples taken. How many cycles are necessary for the estimated effects to approach the effects that would be obtained from an infinitely long chain? In practice, we run several chains and use the average SNP effect from the several chains. The chains are all independent so we will assume that the estimated effects from any one chain can be modelled as the value from an infinitely long chain plus a sampling error. If the correlation between the estimated SNP effects between two independent chains is $r$, then the correlation between the mean SNP effects from $n_C$ chains and the SNP effects from an infinitely long chain is $\sqrt{n_C r / (n_C r - r + 1)}$. Therefore, for example, if $n_C = 5$ and $r = 0.8$, the correlation of the average estimated SNP effects and the long term SNP effects is $r = \sqrt{0.95} = 0.97$.

Convergence can therefore be checked by comparing the correlation of the SNP effects across chains, and for 5 chains, the mean of 10 correlations is produced, as given in Fig. 2. From Fig. 2 it is seen that BR and BR3+ have essentially converged by 1000 iterations, that is their mean chain correlations are greater or equal to 0.99. However, BR3 does not approach those values until 5000 iterations. This suggests that BR3 requires 5 times more iterations than does the standard Bayes R configuration (BR).

**Estimated heritability.** Figure 1e, f gives the estimated heritability for the sets H10 and H30, where heritability is defined by $h^2 = \sigma_g^2 / (\sigma_g^2 + \sigma_e^2)$. When using the H30 data set, all configurations of BR, BR3, and BR3+ give good estimates of the expected value of 0.3 (see Fig. 1f). However, for the H10 data set, which has a lower signal, there is a consistent 1 to 3% overestimation of the expected value of 0.1. This overestimation is a property of the sample not the method as a GREML estimation of the heritability from the same data produced for H10 and H30: 0.129 ($\sigma = 0.011$) and 0.32 ($\sigma = 0.013$) respectively.

**The distribution of SNP effects on phenotype.** The model for the simulation and the analysis assumed the SNP effects were sampled from a mixture of normal distributions so we can assess the ability of BayesR3 to recover this distribution. The log2 of the number of SNPs falling into each of the 4 mixture components is shown in Fig. 3. The number of SNP effects simulated in the 4 distributions were 396,000, 3485, 500, and 15 going from component 1 to 4, respectively. All methods estimate the number of SNPs simulated in each of the 4 distributions with good accuracy, as compared to the expected counts given by the first stacked bar in each plot, and this estimate is reached at all iteration numbers, except for BR3 where at 50 and 100 iterations there appears to be a lower than the expected number of SNPs falling into the fourth distribution with the largest SNP effects (orange band) for both data sets H10 and H30. As expected, just under 396,000 SNPs are seen to have no effect on the trait, Fig. 3.

**Estimated SNP effects.** Figure 4 provides a visual correlation between the true 4000 effects given in Fig. 4a and their estimated SNP effects for the same 4000 variants using the simulated training set (H30) embed within three different genotype densities: 400,000 markers Fig. 4b, 40,000 markers Fig. 4c and only the causal variants Fig. 4d. That is, each case includes the 4000 causal variants but have varying numbers of SNPs, which have no effect on the trait. All the estimated effects were averaged across five chains, with chain length 2000. Regardless of the number of SNPs included in the analysis, the true SNPs with the largest effects were generally recovered as large effects. However, a large negative peak on Chromosome 26 was missed in part because its minor allele frequency was low (0.0025), and in part because of a breed difference between the Holsteins and Jersey, where proportionally there were 4 times as many Holsteins than Jerseys with the minor allele.

The values above Fig. 4b–d, are all Pearson's correlation values. The first value above each of these figures is the correlation between estimated and true breeding values, the 2nd correlation value is between the true 4000 causal SNP effects to their corresponding estimates, while the 3rd value is the correlation across all SNPs within each of the analysis to their true values, Fig. 4b = 400,000 SNPs Fig. 4c = 40,000 SNPs and Fig. 4d = 4000 SNPs. Comparing Fig. 4b, d the effect of including in the analysis of 396,000 SNPs that have no true effect is to reduce the ability of the analysis to detect SNPs with small effects.

**Processing Speed.** The processing speeds for a training set of 20,000 records across 400,000 SNPs are given in Fig. 5a. The time taken depends largely on the number of outer cycles. Consequently, for the same number of total cycles, BR3 with 25 cycles per inner block is 25 times faster than BR. BR3+ with the same number of outer cycles as BR takes approximately the same time.

**Effect of block size on processing speed.** The time for BayesR3 to process 41,925 records across 400,000 SNPs declines rapidly as

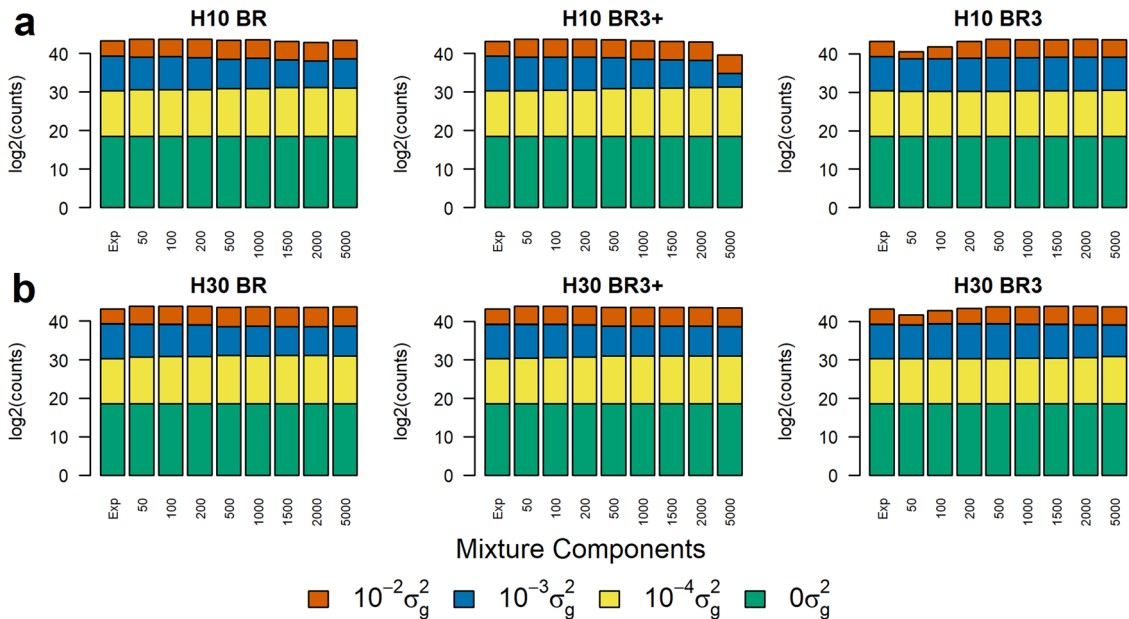

**Fig. 3 Stacked bar plots for the mixture components inferred with respect to BayesR configuration (BR, BR3 and BR3+) and iterations (50 to 2000).** Y-axis is the log2 of the number of SNPs. Component variance $10^{-2}\sigma_g^2$ is given in vermillion, component $10^{-3}\sigma_g^2$ is given in blue, $10^{-4}\sigma_g^2$ is given in yellow and $0\sigma_g^2$ is given in a bluish green colour. The expected (Exp) number of SNPs for each component is given in the first bar in each plot where the expected counts are $(396000, 3485, 500, 15)$. Panel **a** gives results observed for the 10% heritability data set H10, while panel **b** is the equivalent data for the 30% heritability, H30, data set.

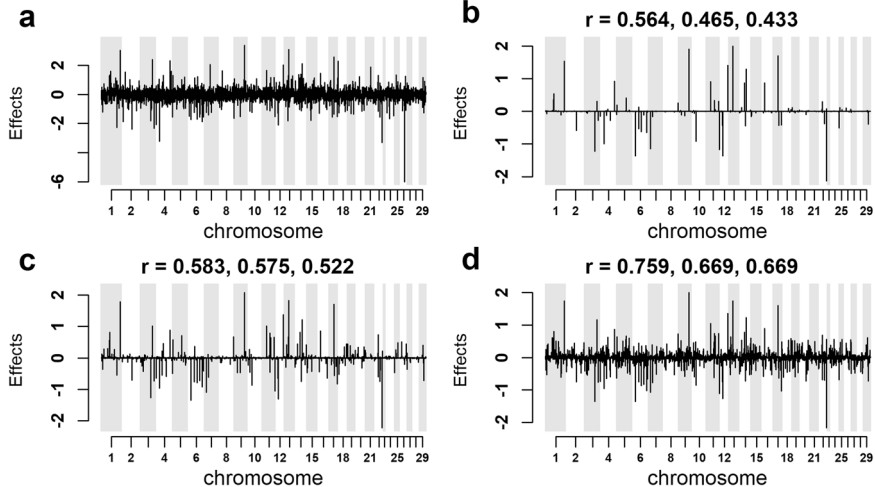

**Fig. 4 The true SNP effects for the 4000 simulated causal variants and their estimated effects using the H30 training data set, embedded within three different genotype densities.** All results are from BR3 using five chains, each of length 2,000 and with a block size of 25. The **r** values above panels **b**–**d**, are all Pearson's correlation values. The first value above each of these figures is the correlation between estimated and true breeding values, the 2nd value is the correlation between the true 4000 causal SNP effect values to their corresponding estimates, while the 3rd value is the estimated SNP effects correlation across all SNPs within each of the analysis to their simulated true values: **a** the simulated true effects for the 4,000 causal variants. **b** the effects of the 4000 causal variants estimated in the training set with 400,000 marker genotypes. **c** the effects recovered from the training set with 40,000 SNP genotypes. **d** the effects recovered when only the true simulated causal variants were used as the genotype set.

block size increases (Fig. 5b). We modelled this curve using the relationship that processing time is proportional to $\frac{n_R + n}{n}$, where $n_R$ is the number of records and $n$ is block size (Fig. 5c). The curves in Fig. 5b, c are almost identical, indicating the model is a good fit. Therefore, for a given number of records we conclude that the processing time for BR3 is proportional to $\left(\frac{n_R + n}{n}\right)$ per SNP, while for BR its speed is expected to be proportional to $n_R$ per SNP. Note also, that the curve in Fig. 5c is scaled to the same range (min and max) given by the curve shown in Fig. 5b. The

accuracies and bias for across-breed genomic predictions associated with selected block sizes from Fig. 5b is given in Fig. 5d. From these results it is seen that from block size 10 to 215 there is an approximately 0.5% drop in accuracy indicating that genomic prediction accuracy, as determined from BayesR3, is robust, within limits, to block size changes. Therefore, we suggest and have determined that block size and the number of inner iterations should be equal, and no greater than, the square root of the number of records (see Supplementary Note 2). For 41,925 records a block size of 205 would be recommended.

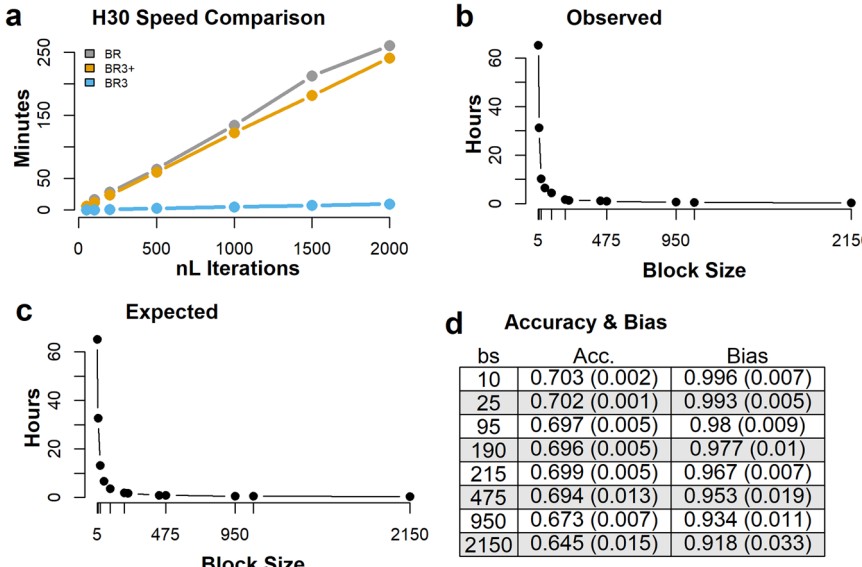

**Fig. 5 Processing speeds for the simulated data sets. a** Y-axis is time in minutes to process 20,000 phenotypic records of 400,000 SNPs for the 3 Bayes R configurations as specified in Fig. 1. **b** Computing time in hours of BR3 with respect to changing block size for Markov chain lengths of 10,000 and for block sizes: $n \in \{5, 10, 25, 50, 95, 190, 215, 430, 475, 950, 1075, 2150\}$, for the simulated data set using 41,925 phenotype records and 400,000 SNPs composed of Holsteins and Jersey cows only, **c** ratio of $\frac{n_R + n}{n}$, for the same block sizes, $n$, and where $n_R$ is the number of records. Note plot in panel **c** is scaled to have the same range (min and max) given in plot **b**. **d** Aussie Reds genomic prediction mean accuracies (**Acc**.) and biases (standard deviations) from 5 MCMC chains of length 10,000 each, for selected block sizes associated to the timings given in **b**.

---

**Table 1 Comparison between BR3C and BR3 for prediction accuracy for the 10% (H10) and 30% (H30) heritability true breeding values simulated data sets.**

| Heritability | Breed (# Records) | BayesR Method | Accuracy ($\sigma$) | Bias ($\sigma$) | nSNP |
|---|---|---|---|---|---|
| 10% (H10) | Holstein (3753) | BR3C | 0.746 (0.002) | 0.936 (0.005) | 3558 |
| | | BR3 | 0.670 (0.003) | 0.964 (0.012) | 3948 |
| | Jersey (1247) | BR3C | 0.722 (0.003) | 0.899 (0.011) | 3558 |
| | | BR3 | 0.640 (0.004) | 0.954 (0.007) | 3948 |
| | RDC (1725) | BR3C | 0.649 (0.003) | 0.976 (0.014) | 3558 |
| | | BR3 | 0.534 (0.024) | 1.073 (0.033) | 3948 |
| 30% (H30) | Holstein (3753) | BR3C | 0.850 (0.001) | 0.990 (0.005) | 3111 |
| | | BR3 | 0.754 (0.003) | 0.998 (0.005) | 4707 |
| | Jersey (1247) | BR3C | 0.809 (0.001) | 0.988 (0.006) | 3111 |
| | | BR3 | 0.683 (0.005) | 0.977 (0.007) | 4707 |
| | RDC (1725) | BR3C | 0.759 (0.004) | 0.961 (0.013) | 3111 |
| | | BR3 | 0.577 (0.011) | 0.949 (0.025) | 4707 |

Prediction accuracy was tested in 3 validation breeds, and the table also shows the average number SNPs (nSNP) included in the model.

---

**BayesRC (BR3C).** BayesR, like most genomic prediction methods, assumes that the prior probability that a marker affects a trait is the same for all markers. To address this limitation BayesRC was introduced by MacLeod et al.[25], to take advantage of biological knowledge associating genes or SNPs to traits. We implemented BayesRC within our BayesR3 framework (see Methods) and here this configuration is identified as BR3C. Then we analysed the simulated data where its 400,000 SNPs were assigned to two classes: class 1 contained the 4000 known causal variants, while class 2 contained the other 396,000 SNPs. The effect on predictions accuracy and bias for six test data sets are seen in Table 1 and depending on breed and heritability BR3C improved the prediction accuracy over BR3. The estimated mixture components for these two classes are given in Table 2. BayesR3C estimates the proportion of class 1 SNPs that affect the trait to be ~20%, whereas it estimates the proportion in class 2 to be 1%. Thus, the analysis does discover the difference between the 2 classes but

**Table 2 The average number of SNPs for the 4 mixture components ($k_1$, $k_2$, $k_3$, $k_4$) retrieved from BayesR3C.**

| Heritability | Class 1 | | | | Class 2 | | | |
|---|---|---|---|---|---|---|---|---|
| | $k_1$ | $k_2$ | $k_3$ | $k_4$ | $k_1$ | $k_2$ | $k_3$ | $k_4$ |
| H10 | 3365 | 60 | 495 | 19 | 392822 | 3060 | 57 | 0.03 |
| H30 | 2900 | 328 | 695 | 16 | 393783 | 2155 | 0.3 | 0.03 |
| Expected | 0 | 3485 | 500 | 15 | 396000 | 0 | 0 | 0 |

Class 1 contains the 4000 casual variants and where the expected counts are seen in the row labelled Expected. Class 2 contains no casual variants. Note, for class 1 the number of SNPs does not add up to the expected 4000 because 61 variants have a minor allele frequency less than 0.002 in this data set, which was the threshold of inclusion into the analysis.

underestimates it (all the SNPs in class 2 should have zero effect). This lack of power is because of the LD between SNPs in the 2 classes and the large number of SNPs in class 2.

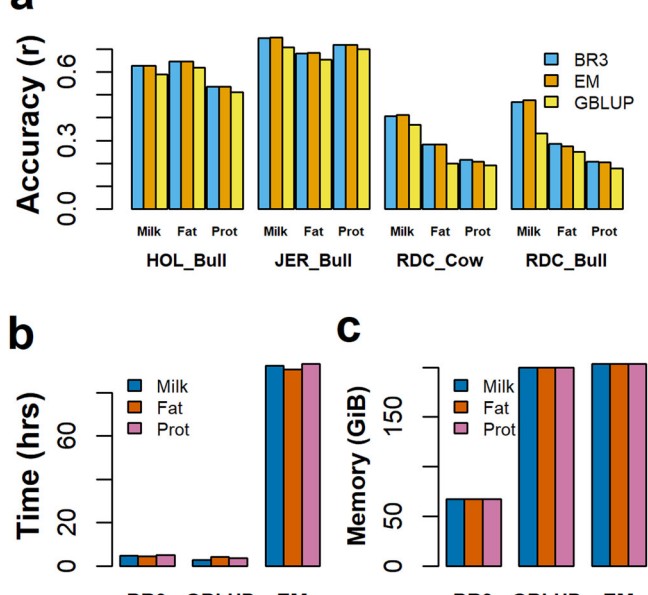

**Fig. 6 Comparison of the accuracy of genomic prediction and computational efficiency between BayesR3 to EM-BayesR (EM) and GBLUP.** Each comparison is for a single trait analysis for milk, fat, and protein yield, using a reference set of 25,000 Holstein and Jersey cattle and where accuracy was tested in 3 validation sets: 398 Jersey bulls, 702 Holstein bulls, and 3082 RDC cows and 212 RDC Bulls. **a** Accuracy of genomic prediction as a function of trait and breed. Results for BR3 are the sky-blue bars, EM-BayesR orange bars, and GBLUP yellow bars. Also note, error bars are not included as GBLUP only provides single-point values. Computation requirements in terms of **b** runtime and **c** memory requirements. Note with respect **b**, **c** Results for milk, fat and protein are given by blue, vermillion, and reddish-purple bars respectively.

### Analysis of milk yield and composition

*Processing speed and memory requirement.* A comparison of BayesR3 to EM-BayesR[15] and to GBLUP (MTG2 software[27]), using 25,000 animals with real phenotype data for milk, fat, and protein yields and 717,463 SNP effects shows that BayesR3 is as accurate as EM-BayesR, but 4.3%, on average, more accurate than a GBLUP (Fig. 6a). These accuracies are the correlation between estimated breeding value and phenotype for bulls that were not included in the training data. Also, BayesR3's average processing time of 4.7 h was more than 19 times faster than the 92.1 h recorded for EM-BayesR, while the average processing time for GBLUP was only 3.6 h (Fig. 6b). BayesR3 required 33% less RAM than either EM-BayesR or MTG2/GBLUP (Fig. 6c).

*Multi-trait analysis of mid-infrared spectra from milk of dairy cattle.* Modern measurement techniques often generate data on many variables. This includes many "omic" technologies. Here BayesR3's multi-trait facility was used to analyse mid-infra-red (MIR) spectrometry on milk samples of 9834 Australian dairy cows. The MIR data consisted of absorption peaks recorded for each cow at 537 wavenumbers[28]. As the traits in the multi-trait model were assumed to be uncorrelated, the wavenumber peaks, which are considered here as phenotypes/traits, were first transformed into PCA components, from which the first 17 components that accounted for 99% of the total variance were chosen for multi-trait analysis. The animals had high-density genotypes at 717,463 SNP. Multi-trait BayesR3, like multi-trait BayesR[24], has two additional variables (as explained in the Material and Methods) describing the mixture distributions. A SNP may be

determined to be excluded from the model (meaning it affects none of the traits) or included in the model in which case it is then independently assigned to one of the four distributions for each trait. Therefore, it is possible but unlikely for a SNP to be included in the model but be assigned to the null distribution for all traits. As with the single-trait analysis, multi-trait BayesR3 processes SNPs in blocks to reduce processing time.

The MIR PC trait summary results are given in Fig. 7, including the observed phenotype variance $\sigma_p^2$ for each MIR PC trait plotted against PC number (Fig. 7a). Note, as expected, PC1 has the largest variance and it reduced to PC17. There was only a slight tendency for $h^2$ to decrease from PC1 to PC17 (Fig. 7b). Therefore, both the genetic and environmental variance decreased proceeding from PC1 to PC17. The average number of SNPs, from 717,463, included in the model per iteration was 3995, although not all of these were in the model for any given PC trait. For instance, for PC1, on average 1165 of these SNPs had a non-zero effect or 2830 SNP had a zero effect. However, the number of SNPs in each mixture component is remarkably similar as seen in Fig. 7c, d. Across the 17 PCs, SNPs on average affect 4 traits even though the PCs are uncorrelated.

The number of SNPs with a posterior probability (pp) of being included in the model > 0.9 was 20 as given in Supplementary Data 1, sheet S3. Some of these SNPs were well known QTL for milk fat% or protein% such as *DGAT1*, the casein gene cluster and *PAEP* (Supplementary Table 1) Other SNP marked more detailed aspects of milk composition such as the SNP on chromosome 1 at 142.8 Mb which was close to the QTL affecting phosphorous, magnesium, potassium, and sodium concentration in milk[29,30]. However, in other cases no one SNP had a pp > 0.9, but multiple closely spaced SNPs' pp summed to >0.9. If several SNPs are in high LD, it is likely that there is insufficient evidence to select one over all the others so the MCMC process will fit one of the SNPs in one iteration and a different SNP in other iterations. After summing the pp across all SNP in each 50 kb window, we found 43 segments with sum(pp) >0.9 (Fig. 8a). The simplest interpretation is that these segments each contain a causal variant affecting the MIR spectra. A large number of such segments is consistent with the estimate of 3995 SNPs associated with the 17 PCs. Some of these 43 segments also correspond to known QTL for milk composition traits such as the QTL for lactose concentration on chromosome 27 at 36 Mb, at 15 Mb on chromosome 3, and on chromosome 5 at 31 Mb[31] (see Supplementary Table 1 & Supplementary Data 1, sheet S7).

*Multi-trait analysis of milk production traits of dairy cattle.* To further demonstrate the use of multi-trait BayesR3 for mapping QTL and to confirm the overlap with MIR QTL, a large data set was analysed of 65,637 dairy cattle recorded for milk, fat, and protein yield (traits converted to PC). The observed mixing proportions and heritabilities are in Supplementary Table 2. Across the 3 traits a total of 9948 SNPs per iteration had an effect. Supplementary Table 2 shows that only 140 to 2688 of these 9948 SNPs had no effect on one of the PC traits and thus most of the 9948 SNPs affected all 3 PCs. Most SNP effects are very small being drawn from the smallest non-zero distribution.

There were 35 individual SNPs with a posterior probability of inclusion in the model (pp) >0.9 (Supplementary Data 1 sheet S2). Again, many of these represented known QTL for milk production traits (e.g., chromosome 3 at 15 Mb and chromosome 5 at 31 Mb) but others had not previously been reported (e.g., chromosome 8 at 12 Mb, chromosome 17 at 0.5 Mb). As with the MIR data, there are many cases where closely linked SNPs each had a moderate pp. There were 556 regions with sum(pp) >0.9 that include many previously reported QTL: for instance, Chromosome 20 at 31 Mb and 58 Mb, spanning the growth hormone receptor gene, *GHR*,

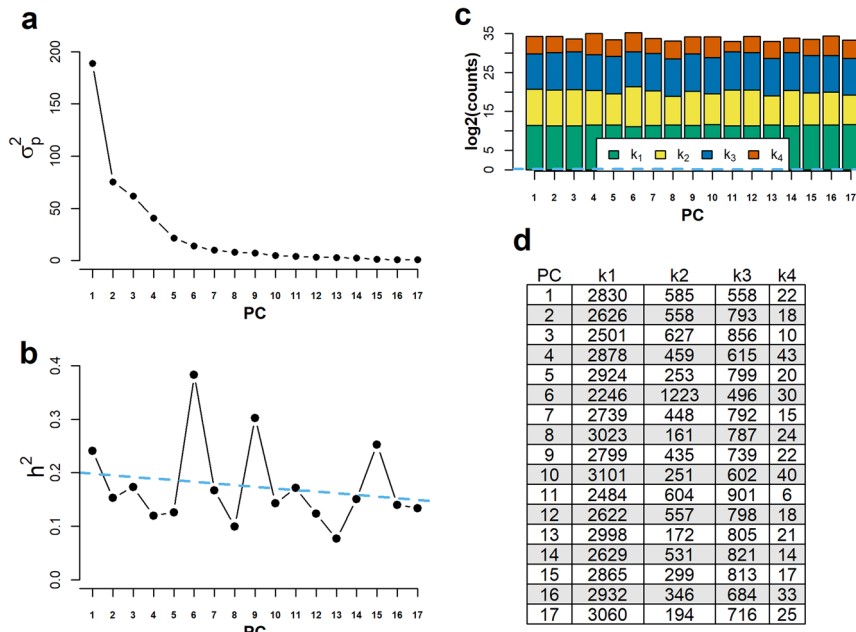

**Fig. 7 MIR PC trait summary results. a** The phenotype variance $\sigma_p^2$ for each MIR PC trait plotted against PC number. **b** Estimated heritability for each PC trait. **c** Number of SNPs per mixture distribution for each PC trait. Note, count $k_4$ is given in vermillion, count $k_3$ is given in blue, $k_2$ is given in yellow and count $k_1$ is given in a bluish green. **d** Raw counts for the number SNPs per distribution for each trait. Note the sum of counts for each PC trait is 3995, which is the number SNPs estimated to be associated to the traits.

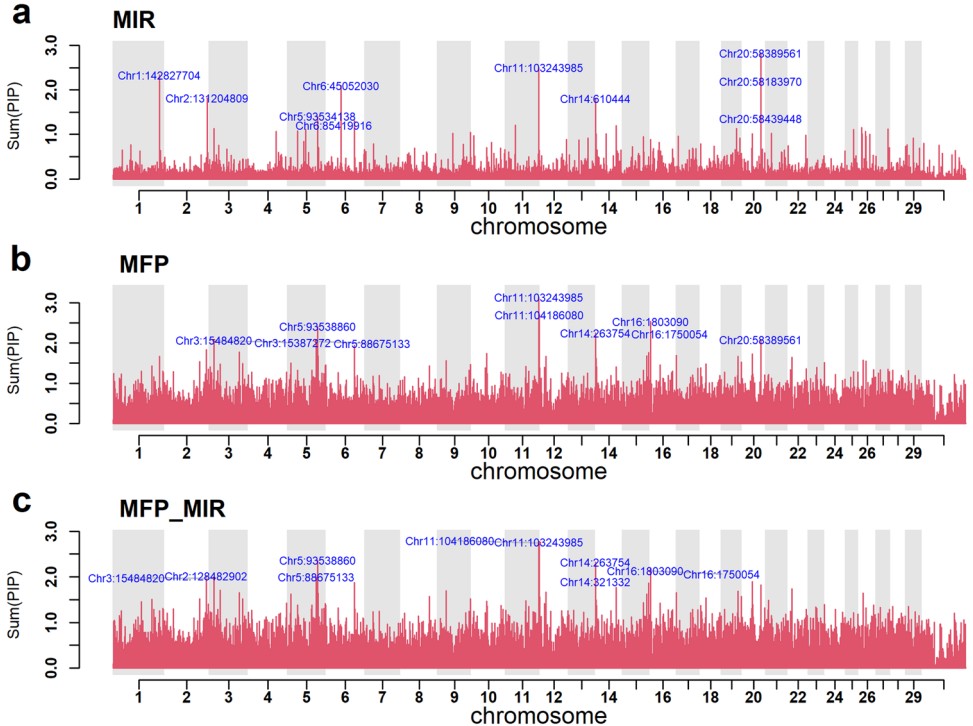

**Fig. 8 Manhattan plots for MFP, MIR, and MFP_MIR multi-trait analysis.** Y-axis is the sum of the posterior probabilities that SNPs within and centred on each non-overlapping 50 kb segment of the genome is included in the model. **a** Result from the 17 PCA MIR multi-trait analysis. **b** Plot is multi-trait milk, fat, and protein yields. **c** Multi-trait milk, fat, and protein yield analysis, using BayesR3C, where class 1 was formed from the top 1000 SNPs identified from the MIR analysis. Note each plot has the top 10 SNP effects labelled.

and the inorganic pyrophosphate transport regulator, *ANKH* (Fig. 8a and Supplementary Table 1).

Although the analysis of the MIR data and the milk yield data used different animals as well as different phenotypes, they identified many of the same SNPs/QTL regions (Fig. 8a, b). Of the 1000 SNPs with the highest posterior probability, 38 are included in both lists when only 1 would be expected by chance (Supplementary Data 1, sheet S5). Therefore, we use the MIR

results as prior information in a BayesR3C analysis of the same milk production data (Supplementary f Our Statistical Analysis sections has been renamed as Statistics and Reproducibility Data 1, sheet S3).

BayesR3C was used to re-analyse the milk production data, but this time the SNPs were divided into 4 classes based on our prior knowledge of the posterior probability that they were associated with milk composition as estimated by the analysis of the MIR data. Classes 1 to 4 were in declining order of the MIR posterior probability. Supplementary Table 3 shows the mixing proportions within each of the 4 classes. In class 1, which contains the 992 SNPs with the highest posterior probability to affect MIR traits, 4.8% of the SNPs had a non-zero effect on milk, fat, or protein yields, whereas in the other 3 classes only 1.6% of the SNPs influenced milk, fat, or protein yield. In addition, the proportion of SNPs with a larger effect (distributions 3 and 4) declined between class 1 to 4. Thus, SNPs with an effect on milk MIR spectra are more likely to affect milk, fat, or protein yield and more likely to have a large effect than SNPs that do not affect MIR spectra. Using the MIR prior information increases the probability that some SNPs are included in the model for milk, fat, and protein yield. The number of SNP with pp > 0.9 increased from 35 to 40 (Fig. 8c). A comparison of BayesR3C using the MIR prior and BayesR3 for MFP and MIR revealed that some QTL overlapped between methods, such as chromosome 11 at 103 Mb near *PAEP* (see Supplementary Table 1). Note also that the number of segments with pp > 0.9 increased ($N = 627$) compared to the BayesR3 ($N = 556$ 50-kb segments) analysis. The y-axis in Fig. 8 shows the number of SNPs independently associated with the traits in each 50 kb interval. If there was a single causal variant in a 50 kb segment this value might be expected to be 1.0. However, it is sometimes more than 1.0 due to a combination of multiple causal variants, more than 1 SNP needed to explain a single causal variant, and sampling error.

## Discussion

BayesR can be used for 3 purposes—it can describe the genetic architecture of a complex trait; it can map SNPs associated with the trait and it can predict the genetic value or breeding value or individual risk from an individual's genotypes at markers such as SNPs. BayesR assumes a flexible distribution for SNP effects: it can accommodate many SNPs with no effect, some with a small effect and a very few with a large effect and this is the pattern we have found in this analysis and elsewhere[32]. The more flexible prior usually results in higher prediction accuracy than methods that assume all SNP effects are drawn from a single normal distribution such as GBLUP. However, Bayesian methods that are implemented by MCMC require longer computer times than BLUP methods. This paper describes a faster method for implementing BayesR. The increase in speed is achieved by updating the SNP effects in blocks and cycling through the SNPs within a block several times before moving to the next block. That is, there are inner cycles within a block and outer cycles among the blocks.

The most common method for implementing these Bayesian models is to update one SNP, then correct the residual phenotype for all individuals and proceed to the next SNP. Thus, in every cycle for every SNP a pass is made through the list of individuals causing the computer time to be proportional to $C \times R \times M$, where C is the number of cycles, M is the number of markers and R is the number of phenotypic records. An alternative[22] is to store the $V'V$ matrix and $V'y$ vector in memory. This avoids processing all individuals for each SNP, but at the cost of a large matrix multiplication for every SNP. Consequently, this latter approach is not advantageous when the number of SNPs is large. The method

proposed here is in between these two approaches—we form $V'_b V_b$ and $V'_b y$ for a block of SNPs and cycle around this block $n$ times, which is also equal to the number of SNPs in the block, before returning to the outer loop by updating the residuals and moving to the next block.

Calus[19] also used a method that divided the SNPs into blocks but with two important differences: They used blocks of approximately 4 to 6 SNPs depending on the number of records being processed, whereas we use blocks of size equal to approximately $\sqrt{n_R}$, where $n_R$ is the number of records and they did only one cycle while a block was in core. They achieved a reduction in time needed by coding a block of about 4 to 6 SNP genotypes into a single number. This reduced the time necessary to process the data on all individuals and appears to be competitive with the EM-Hybrid version of BayesR[15]. However, we found without hardware support, that storing 4–6 SNP genotypes as a single number offered little speed advantage compared to modern matrix routines with built-in vectorization.

The simulation results show that BayesR3 can estimate the proportion of variance explained by the SNPs, the number of SNPs in each distribution, and the position of the causal variants. The multiple trait version was used to analyse empirical milk MIR spectra data represented by their first 17 PCs. Although the 17 PCs were uncorrelated, they are largely explained by the same SNPs, meaning that each MIR trait is mostly associated with the same set of SNPs.

On average 3995 SNPs were included in the model and about 1000 of these influenced any given PC trait. The SNPs with a high posterior probability of being included in the model are presumably tracking causal variants with which they are in LD. 20 SNPs had a posterior probability >0.9 and many are close to known polymorphisms (QTL) affecting milk composition. The use of 17 PCs captures information on more specific traits, such as lactose concentration, in that QTL affecting lactose concentration, phosphorus, and other milk mineral concentrations were also detected by the analysis.

The multi-trait BayesR3 on milk, fat, and protein yields, found that 35 SNPs were included in the model on >90% of MCMC cycles implying there may be a causal variant linked to these SNPs. However, often the signal from a causal variant is split among closely linked SNPs. Therefore, we calculated the sum of posterior probabilities for SNPs within a 50 kb segment and showed that 556 of such segments had sum(pp) > 0.9, indicating we mapped 556 QTL for milk, fat, and protein yields to within 50 kb segments of the genome.

As expected, we found considerable overlap between the SNPs that affected milk, fat, and protein and those that affected the milk MIR spectra. We used the MIR data to help map the variants causing variation in milk, fat, and protein using BayesR3C, which allows different mixing proportions among classes of SNPs. In this case, we analysed milk, fat, and protein traits but defined classes based on whether the SNP affected the MIR PCs. The 992 SNPs with the largest posterior probability to affect MIR traits were enriched for effects on milk, fat, and protein yield and for large effects on these 3 traits. For instance, of the 992 SNPs with biggest effect on MIR, 48 SNPs per iteration were included in the model for milk, fat, and protein. However, among the other 634,879 SNPs, 10,810 were included in the model for milk, fat, and protein. That is, the 992 SNPs with the biggest effects on MIR spectra comprise only 0.4% of the total SNPs affecting milk, fat, and protein. This may illustrate a common problem with using "omic" traits to identify variants affecting target complex traits: the individual "omics" datatypes detect only a minority of the variants affecting the target traits.

In the simulated data, BayesR3C achieved much higher accuracy in predicting genetic value of individuals than BayesR3. In this data the classes defined were extremely different in the

proportion of causal variants contained, that is class 1 contained all the causal variants. However, in the milk, fat, and protein yield data BayesR3C did not achieve a higher accuracy than BayesR3 (see Supplementary Table 4), probably because the four classes defined from the MIR data did not differ enough in their effect on milk, fat, and protein yield. However, there was an increase in 8 SNPs with pp > 0.9 (35 to 43) and the number of 50 kb segments with 90% probability of containing a causal variant increased from 556 with BR3 to 627 with BR3C.

A limitation of the multi-trait BayesR model is that it assumes that the genetic and residual covariance matrices are proportional to each other. This is often a good approximation of the observed covariances, and it has the advantage of avoiding the use of poorly estimated covariance matrices which often occur when many traits are considered. To make the computation easier we transform the traits to uncorrelated traits. However, despite using uncorrelated traits, the model estimates whether the same SNPs can explain variation is multiple traits. This is a major advantage over single trait analysis which often finds two closely linked SNPs, one affecting one trait and the other another trait. Typically, we would like to know if one SNP could be used for both traits and the multi-trait model does this. Nevertheless, a future development of the method should overcome the need to assume this proportionality of covariance matrices.

In conclusion, BayesR estimates the number of polymorphisms affecting a trait and the distribution of their effect sizes, maps the position of these polymorphisms on the genome, and predicts the genetic value of individuals with genotypes but no phenotypic measurement. The methodology (BR3) decreases substantially the computer time needed for regressing phenotypes against genotypes such that it is now comparable with a GBLUP analysis. By applying BR3 to milk yield and composition data, we estimate that approximately 10,000 high-density SNPs are independently associated with these traits and for 627 of them we map the causal variant to within 50 kb. BR3 can also analyse high dimensionality 'omic' data and can incorporate prior biological knowledge about the genome variants used. Using these facilities, we show that milk MIR spectra identify many variants that affect milk, fat, and protein yield, but there are 1000's of variants affecting milk, fat, and protein yield that were not detected by the MIR spectral data.

## Methods

### Data

*Simulated data.* Phenotypes were simulated using 64,345 real cattle genotypes by randomly selecting a set of 4000 genome-wide SNP as causal variants (QTN). The 4000 QTNs were chosen from a set of 983,116 imputed autosomal sequence variants. The sequence variants were imputed using Run5 of the 1000 Bull Genomes Project[33] and one of a pair were pruned for high linkage disequilibrium (LD $r^2 > 0.9$ in 0.5 Mb windows). Variants with MAF < 0.002 were removed, intergenic variants were reduced to the set overlapping the High-Density Illumina BovineHD 800 K panel (162,583), and half of the remaining variants within and close to genes (up to 5000 bp from gene start and end positions) were randomly removed leaving the final set of 983,116 variants. The QTN effects and trait phenotypes were simulated using the approach described in ref. [25]. Briefly, QTN effects were simulated by sampling 15, 500, and 3485 effects from each of 3 normal distributions with variance $0.01\sigma_g^2$, $0.001\sigma_g^2$, and $0.0001\sigma_g^2$ respectively, and where $\sigma_g^2$ is the additive genetic variance (set at 625). The genetic value or true breeding value (TBV) of each animal was then calculated as: $\text{TBV}_i = \sum_{j=1}^{4000} v_{ij}\varepsilon_j$ where $\varepsilon_j$ is the $j$th QTL effect and $v_{ij}$ represents the $j$th genotype (coded 0, 1, or 2 for genotypes aa, Aa, and AA) for record $i$.

An environmental effect for each animal was sampled from a normal distribution and was added to the genetic value to produce a phenotypic trait record with a heritability ($h^2$) of either 0.1 ("H10") or 0.3 ("H30"). The same 4000 QTN positions were used for both trait heritabilities and a new set of QTN effects were simulated for each trait. The cattle were mainly purebred Holstein, with some purebred Jersey and Australian Reds (RDC). All 1725 RDC were used as a validation set, while Holstein and Jersey animals were allocated to the training populations. From this simulated data, we randomly selected two training sets for

analyses in this study: one with 41,925 Holstein and Jerseys and a second set of 20,000. From the 983,116 SNP genotypes available for each animal, we randomly selected three subsets of genotypes from the SNP map file, but also included the 4000 QTN: 400,000, 40,000, and 4000 SNP. Thus, the latter set represents the QTN only.

*Empirical milk production data—multi-trait analysis.* A training population of 65,637 dairy cows and bulls was available with milk, fat, and protein yield phenotypes (MY, PY, and FY, respectively) where the raw data was pre-corrected for known fixed effects (herd, year, season, and lactation). The bull data was derived from daughter records. The cattle included Holstein (80%), Jersey (15%), and crossbreds (5%). All animals were genotyped for either low (~9000 SNP), medium (~50 K SNP), or high density (HD 800 K SNP) panels. The low-density panels mainly overlapped the 50 K panel but also included a custom set of ~ 1000 variants that have been found to be more predictive than random markers for milk and other traits. Animals with low-density genotypes were imputed to 50 K (imputation reference of 14,722 animals) and all 50 K genotypes were imputed to the HD panel (imputation reference of 2700 animals). All markers not overlapping either the HD or 50 K panel were also imputed for all individuals to generate a final genotype set of 717,463 SNP. All SNP markers were mapped to the ARS-UCD1.2 reference genome[34] and imputation was done using FImpute and default parameters[35]. This set of phenotypes and genotypes was used for a multi-trait analysis to fine-map candidate genes for these traits. The phenotypes were converted to three principal component traits.

*Milk production data for methods comparison.* A subset of 25,000 animals of the above set of 65,637 (purebred Holstein and Jersey only) were used as a training set to run a comparison of computational efficiency and accuracy of the BayesR3 to the previously published EM-BayesR[15] and a GBLUP (Genomic Best Linear Unbiased Prediction) approach using MTG2[27]. The reference set composition was 13,413 Holstein cows, 6623 Jersey cows, 4,289 Holstein bulls, 675 Jersey bulls. Accuracy was tested using 3 validation sets: 398 Jersey bulls, 702 Holstein bulls, and 3082 RDC cows. Any sires/dams or sons/daughters of the validation animals were removed from the training set.

*MIR data.* Milk samples of 9834 Australian Holstein (83%), Jersey (9%), and crossbred cows (8%) calved in Spring 2017 from 21 commercial herds were taken (2 to 8 times/cow) and analysed for milk composition by an infrared spectrometer (Model 2000, Bentley Instruments, Chaska, MN) and the corresponding spectra were stored for this study. Each single spectrum included 899 data points, with each point representing the absorption of infrared light through the milk sample at wavelengths from 649 to 3999 cm$^{-1}$ regions. Several mathematical treatments were applied to the raw spectra, including removal of noisy and uninformative area, eliminating outliers, using a standardized Mahalanobis distance calculated with the 'mahalanobis' function of the 'stats' R package (R Development Core Team, 2020), and taking the first derivative of the MIR signal using the 'gapDer' R function[36], as commonly recommended for MIR studies[28]. This resulted in 537 wavenumbers per sample. Each cow supplied on average 3.7 milk samples; therefore, multiple spectra were averaged resulting in a single spectrum being assigned to each cow for the analysis. These wavenumbers were then corrected for Herd ID, and age of calving, leaving only breed (3 levels) and the mean as the fixed effects used in the BayesR animal genetic model for analysis. The genotype set assigned to each cow was the same 717,463 SNP set as explained for the milk production data above. Note the 9834 cows used in the MIR analysis are separate from the 65,637 dairy cows used in milk production data analysis described above.

## Statistics and reproducibility

*The genotype matrix V.* The elements of $V$ were scaled and centred: $v_{ij} = \frac{v_{ij}^* - 2p_j}{\sqrt{2p_j(1-p_j)}}$, where $v_{ij}^*$ is the genotype of animal $i$ at SNP $j$. The genotypes are represented by 0 for one homozygote, 1 for heterozygote and 2 for the other homozygote, $p_j = \sum_i v_{ij}^*/(2n_R)$ gives the observed allele frequency for SNP $j$ in the $n_R$ records. Note, SNP $j$ is considered monomorphic if its Minor Allelic Frequency (MAF) is less than 0.002 and it is excluded from the analysis.

*The statistical model (single trait).* BayesR and Gibbs sampling was used to implement the mixed effects model: $y = Xu + Vg + Za + e$ where $y$ is an $n_R \times 1$ column vector of phenotype/records values, where $n_R$ is the number of records; $X$ is a ($n_R \times n_F$) incidence matrix, $u$ is a $n_F \times 1$ vector of fixed effects and $n_F$ is the number of fixed effects. $V$ is a coded genotype ($n_R \times n_M$) matrix, as constructed above, representing the observed genotypes of each individual across $n_M$ markers; $g$ is a vector containing the SNP effects, $Z$ is an identity matrix ($n_R \times n_R$) if all animals have phenotypes but could be more general if not all animals have a phenotype and $a$ is a vector of random genetic effects not explained by the SNPs with additive variance represented as $\sigma_a^2$. Note, that $e \sim N(0, W^{-1}\sigma_e^2)$ and that $a \sim N(0, A\sigma_a^2)$.

In BayesR the SNP effects are modelled by a mixture of four normal distributions all with zero mean and with zero, very small, small to moderate

variances, respectively. The prior distributions for the BayesR parameters are given as:

$$g_j | \sigma_j^2 \sim N(0, \sigma_k^2)$$

$$
\sigma_k^2 =
\begin{cases}
0 \times \sigma_g^2 \text{ with probability } \pi_1 \\
0.0001 \times \sigma_g^2 \text{ with probability } \pi_2 \\
0.001 \times \sigma_g^2 \text{ with probability } \pi_3 \\
0.01 \times \sigma_g^2 \text{ with probability } \pi_4
\end{cases}
$$

$$\sigma_g^2 = h_t^2 \sigma_t^2$$

$$\pi \sim \text{Dirichlet}(\alpha)$$

$$e \sim N(0, W^{-1}\sigma_e^2)$$

$$a \sim N(0, A\sigma_a^2)$$

where $\alpha = \{1, 1, 1, 1\}$ represents an uninformative Dirichlet prior, $\sigma_e^2$ is the residual variance, $A$ is the additive relationship matrix, $\sigma_a^2$ is the polygenic variance, and $h_t^2$ is the trait heritability. The priors for $\sigma_a^2$ and $\sigma_e^2$ are scaled inverse chi-squares with uniform priors[37]. The trait variance $\sigma_t^2$ is the estimated trait variance obtained from the data, and its exact form is given below (see Material and Methods: Weighted Analysis). Note, $\sigma_j^2$ is the SNP variance which is sampled from the $k^{th}$ mixture distribution that the $j$th SNP is in.

BayesR3 can also include prior biological knowledge about the variants being analysed as done in BayesRC[25]. It does this by dividing the variants into classes based on this prior information. Then the mixing proportions that describe the distribution of variant effects is estimated separately for each class. When there is only 1 class the model is that for standard BayesR.

*Weighted analysis.* In terms of a multi-trait model the residual variance for the $t$th trait is given by $Var(e_t) = W_t^{-1}\sigma_{e,t}^2$, where $W_t$ is a diagonal matrix of weights for trait $t$ and $e_t$ are its residuals. If a record $i$ is missing on trait, $t$ then $W_{i,t} = 0$. If there are no record defined weights, or missing values then $W = I$ the identity matrix. The weighted mean of each trait was determined from $\bar{x}_t = \frac{\sum_i^{n_r} (y_{i,t} * w_{i,t})}{\sum_i^{n_r} w_{i,t}}$ where $w_{i,t}$ is the $i^{th}$ diagonal element of $W_t$ and representing the $i$th record. Therefore, an initial estimate of the phenotypic variance, $\sigma_t^2$, for each trait was

obtained from: $\sigma_t^2 = \frac{\sum \left( \frac{(y_{i,t} - \bar{x}_t)^2}{h_t^2 + \frac{1-h_t^2}{w_{i,t}}} \right)}{n_W - 1}$; $\forall i, w_{i,t} \neq 0$. Note $n_W$ is the number of non-zero weights, and $\sigma_t^2$ resolves to the raw phenotypic variance when $w = 1$. Here, $h_t^2 \in [0, 1]$, represents the heritability for trait $t$ and is a user-specified value, which can be a best guess.

**Sampling**

*Blocked Gibbs Sampling with Inner and Outer Iterations.* We broke the $V_g$ component of the genetic modelled, Eq. (1), into blocks as shown in Eq. (3). Then Gibbs sampling was used to sample the effects in each block $n$-times, where $n$ is the number of makers in the block, before the next block is processed. Then after $m$ iterations across all blocks, Markov chains of length $n_L = m \times n$ were created for each effect. By default, the number of SNPs per block and the number of cycles within a block are the same. This relationship was initially obtained empirically but was also derived from determining the optimal block size as given in the Supplementary Note 2. Also note, that for smaller last blocks they too must undergo $n$ inner cycles to ensure all SNPs are sampled $n_L$ times.

Let $l$ be an iterator from $[1 : m]$. Then at each $l$ iteration (outer iteration), the effects for each block are determined sequentially. On block entry, the right-hand sides for the block $r_b$ are set up by:

$$r_b = V_b' We$$

where $W$ is a diagonal weight matrix and $e$ is the current state of the residuals in the model, Eq. (1). Gibbs sampling on the following conditional posterior distributions with residual updating was used to update the SNP effects within a block, $b$, at each inner iteration $i \in \{1, 2, \ldots, n\}$;

$$g_{bj}^{i+1} \sim N\left( \frac{r_{bj} + (V_b' WV_b)_{jj} g_{bj}^i}{(V_b' WV_b)_{jj} + \kappa I}, \frac{\sigma_e^2}{(V_b' WV_b)_{jj} + \kappa I} \right)$$

$$r_b \leftarrow r_b + (V_b' WV_b)_{.j} \left( g_{bj}^i - g_{bj}^{i+1} \right)$$

where $g_{bj}^i$ represents the $j$th SNP effect, at iteration $i$, $n \geq 1$ is the number of inner iterations, $r_{bj}$ is the $j$th element of $r_b$ and, $n \geq 1$ is the block size and the number of markers. Note, $(X)_{jj}$ represents the $j$th diagonal element of matrix $X$, and that $(X)_{.j}$

is its $j$th column. Also, $\kappa = \sigma_e^2 / \sigma_k^2$ for a BayesR3 solution. Note, the notation $x \leftarrow x + 3$ signifies that variable $x$ is replaced by the value of $x + 3$.

Prior to each update of $g_{bj}^{i+1}$ $k$, a latent indicator variable, is sampled from the data using the following multinomial distribution:

$k = \min_{k'} \{k' \in \{1, \ldots, 4\} : (\sum_{i=1}^{k'} p_i^*) \geq x\}$, and where $x$ is a sample drawn from a uniform distribution: $x \in [0, 1)$. Here $p_k^* = \frac{p_k}{\sum_k p_k}$, where $p_k = \frac{\pi_k}{\left( e^{0.5 * g_j^2 / v_k} \right) \sqrt{v_k}}$, $v_k = \sigma_k^2 + \sigma_e^2$ and $\bar{g}_j = \frac{r_{bj} + (V_b' WV_b)_{jj} g_{bj}^i}{(V_b' WV_b)_{jj}}$, which has the form of a least-squares estimator. When $k = 1$ the SNP effect becomes zero until it is updated again; that is $g_{bj}^{i+1} = 0$ otherwise $g_{bj}^{i+1}$ is set as given above. Then, at the end of the outer iteration and after $\sigma_g^2$ is sampled, $\sigma_k^2$ is updated via: $\sigma_k^2 = \alpha_k * \sigma_g^2$ for the next outer iteration and the mixing proportions are sampled: $\pi \sim \text{Dirichlet}(\alpha + \beta)$. Where $\beta = (\beta_1, \beta_2, \beta_3, \beta_4)$, holds the number of SNPs in each distribution at the end of each out cycle. The SNP markers that get assigned a $k > 1$, at a particular iteration, are said to be in the model, while those with $k = 1$ are said to be out of the model.

On block exit, and prior to any further sampling for any other effects or blocks, the errors $e$ are updated via:

$$e \leftarrow e - V_b'(g_b^{l+1} - g_b^l)$$

where $g_b^{l+1}$ represents the updated block effect after $n$ inner-cycles of the above Gibbs sampler and $g_b^l$ represents the block effect prior to updating. After all blocks are processed once, and at the end of outer iteration $l$, the variances $\sigma_g^2$ and $\sigma_e^2$ get updated from scaled inverse Chi-square distributions as explained below.

*Gibbs sampling of the fixed effects* u. Let $f \in \{1, \cdots, n_F\}$ represent a fixed effect from $n_F$ effects and let $l \in \{1, \cdots, n_I\}$ represent the iteration number and $n_I$ the number of iterations. Set: $e^0 = y$, $u^0 = 0$, and $\sigma_e^2 = \sigma_t^2 * (1 - h_t^2)$. Then at each iteration the fixed effects were sampled by: $\forall f : u_f^{l+1} \sim N\left( \frac{X_f' We^l + X_f' WX_f u_f^l}{X_f' WX_f}, \frac{\sigma_e^2}{X_f' WX_f} \right)$ and $e^{l+1} = e^l - X_f \left( u_f^{l+1} - u_f^l \right)$. Note that $X_f$ represents the $f^{th}$ column of $X$.

*Gibbs sampling of the genetic effects* a. Although not used in any of the analysis presented here, the sampling of the polygenic genetic effects is considered only for completeness and is recommended when the SNPs do not explain all the genetic variance or when it is desired to fit all the SNPs with small effects as well as a small number of SNPs with larger effects. In the former case one can use the A matrix derived from the pedigree and in the latter case a GRM based on all the SNPs[38].

After estimates of g are made the genetic effects a are updated. $Z$ is accompanied by a genomic relationship matrix often extracted from the pedigree known as the A-matrix or a GRM, both of which are used to account for genetic relationships not accounted for by the SNPs.

Let $a^0 = 0$, $\lambda = \sigma_e^2 / \sigma_a^2$. Then at each iteration $l$ and

$\forall i : a_i^{l+1} \sim N\left( \frac{(Z'We)_i - (A^{-1}\lambda a)_i}{(Z'WZ + A^{-1}\lambda)_{ii}} + a_i^l, \frac{\sigma_e^2}{(Z'WZ + A^{-1}\lambda)_{ii}} \right)$ and $e^{l+1} = e^l - Z_i \left( a_i^{l+1} - a_i^l \right)$ and $\lambda = \sigma_e^2 / \sigma_a^2$. Note the notation $(H)_i$ is used to represent the $i^{th}$ element of vector $H$, while $(H)_{ii}$ represents the $i$th diagonal element of matrix $H$. For the starting and initialization, we partition $\sigma_t^2 * h_t^2$ using a 1:9 ratio between $\sigma_{a,t}^2$ and $\sigma_{g,t}^2$ respectively to produce a starting value for $\sigma_{a,t}^2$. Hence, $a^0 = 0$, $\sigma_{a,t}^2 = \sigma_t^2 * h_t^2 * 0.1$ and $\sigma_{g,t}^2 = \sigma_t^2 * h_t^2 * 0.9$.

*Sampling the variances.* After each Gibbs iteration, and according to a sampling schedule (see below) the variances were sampled using inverse scaled Chi-square distributions via: $\sigma_e^2 \sim \frac{\sum_i (e_i^2 W_{ii})}{\chi^2(n_w - 2)}$, $\sigma_a^2 \sim \frac{a'G^{-1}a}{\chi^2(n_r - 2)}$ and $\sigma_g^2 \sim \frac{n_m g'g}{\chi^2(n_m - 2)}$. Where $\chi^2(x)$ is a Chi-square distribution with $x$ degrees of freedom; $n_w$ is the number of non-zero weights and $n_r$ is the number of records associated with the specified trait, and $n_m$ is the number of SNPs in the model for this iteration, see also[26]. That is, $n_m = \sum_{k=2}^4 \beta_k$. After which, all $\sigma_k^2$ are recalculated as given above.

*Scheduling sample.* To ensure all model effects are sampled within the block scheme the same number of times as the SNP effects, a sampling scheduling was set up for scheduling the sampling of these extra effects and variances, by interlacing their sampling within the outer cycles. The number of outer cycles ($m$) is determined from $m = n_L / n$, where $n_L$ is the total number of samples to be drawn and therefore, $n_L$ represents the Markov chain length. The rate of the extra sampling events is determined by $s_r = \max(\frac{n_M}{n^2}, 1)$, where $n_M$ is the number of SNPs and $s_r$ is the sampling rate, and to that end the block size ideally should be in the range $0 < n \leq \sqrt{n_M}$. Then after every $s_r$ blocks are processed, the sampling of fixed and pedigree effects and the sampling of the variance $\sigma_e^2$ occurs, while the sampling of the variance $\sigma_a^2$ and the Dirichlet parameters must occur at the end of each outer cycle because they require all SNP effects to be sampled first.

**Multi-trait model and analysis.** The multi-trait model for BayesR, was presented by Kemper et al.[24] but we will describe it briefly for completeness. The model for

each trait is the same as described for the single trait case. That is using subscript $t$ to indicate trait $y_t = Xu_t + Vg_t + Za_t + e_t$. It is assumed that the traits are uncorrelated or decorrelated, by being replaced by their respective PCA components, or Cholesky transformed traits[24,39]. However, the multi-trait model includes an indicator variable $J$. When $J = 0$, the current SNP under consideration has zero effect on all traits. However, if $J = 1$ then the SNP can be associated with one or more traits, and in this case the SNP effect is sampled for each trait independently. Therefore, there is a vector $\pi$ for each trait which stores the probabilities that $k = 1$, 2, 3, or 4 for that trait. However, the interpretation of $\pi$ is slightly different to the single trait case because it is the probability conditional on $J = 1$. In BayesR, when a SNP is processed, first $J$ is sampled based on all traits and if $J = 1$ the effect on each trait is sampled independently as in the single trait case.

The multi-trait model requires an additional parameter $\pi^* \sim \text{Beta}(\alpha^*)$, for the probability that $J = 1$. For the current SNP $J$ is sampled $J = 1$ with probability $p$ and $J = 0$ with probability $1 - p$ where (using subscript t for traits and k for the component of the mixture within each trait):

$$p = \frac{1}{1 + e^{\left(p(\bar{g}|J=0) - p(\bar{g}|J=1)\right)}}$$

$$p(\bar{g}|J=0) = \log(\pi_0^*) + \sum_t \log\left(L_{t,0}\right)$$

$$p(\bar{g}|J=1) = \log(\pi_1^*) + \sum_t \log\left(\sum_k L_{t,k} * \pi_{t,k}\right)$$

$$L_{t,k} = \sqrt{v_{t,k}} * e^{-0.5*v_{t,k}*\bar{g}_{j,t}^2}$$

$$v_{t,k} = \frac{1}{\left(\sigma_{k_t}^2 + \sigma_{e_t}^2\right)}$$

where $\bar{g}$ is the least square estimate of the effect as specified previously and $\bar{g}^2$ is its square. $L_{t,k}$ is the likelihood for trait $t$ of SNP $j$ being sampled from distribution $k$. If $J = 1$, then the sampling of the effect of each trait was as described for the single trait case and each trait was processed independently.

The probabilities that a SNP is in the model ($\pi^*$) is sampled at the end of each outer iteration using: $\pi^* \sim \text{Beta}(\alpha^* + \beta^*)$. where $\alpha^* = \{1, 1\}$ and $\beta^* = \{\beta_1^*, \beta_2^*\}$ where $\beta_1^*$ is the number of SNPs currently not in the model and $\beta_2^*$ is the number of SNPs currently in the model. The initial values, for the first iteration were set to $\pi^* = \{0.95, 0.05\}$.

**Comparative analysis: BayesR3, EM-BayesR and GBLUP**. The computational efficiency of BayesR3 was compared to a GBLUP approach as well as the previously published fast version of BayesR, referred to as EM-BayesR[15]. For each approach, we ran the equivalent univariate models for three milk trait phenotypes using the 25,000-animal training set described above. The animals were genotyped for 717,463 SNP and 87,698 of those with minor allele frequency <0.005 were not included in the analyses. The GBLUP approach was implemented using MTG2 software[27], and this requires the genomic relationship matrix (GRM) to be pre-calculated prior to running GBLUP so we first calculated the GRM using GCTA software[40] invoking the multi-threading option.

The EM-BayesR approach was implemented as described in ref. [15] using in house software written in C++. The EM part of the analysis completed either when convergence of 1e-7 was reached or a maximum of 1500 iterations, whichever came first. The BayesR part of the analysis was set to run for 4000 iterations. For both the EM-BayesR and the BayesR3, five MCMC chains were run in parallel to check that the results had stabilised across chains. BayesR3 was run for 40,000 iterations, with 20,000 as burn-in. The accuracy of prediction was calculated as the correlation between the predicted and known phenotypes for each of the three validation sets. The accuracy and bias were calculated for each chain and then averaged.

The analyses for all three methods were run on a Linux High Performance Computer cluster (described below) using Intel® Xeon® Platinum 8168 Processors (2.70 GHz), each with 24 cores, 48 CPU, and 740 Gb memory.

**BayesR3 software**. The software is written in C++ using the Eigen (version 3.4.0) C++ template library for linear algebra. It has been compiled for the Linux platform using Intel's compiler ICPC with the Eigen's Intel MKL and openMP library options turned on, and it has also been built for the Windows 10 platform using Visual Studio Community (2022) edition of C++ version 17.1.6

**Computing specifications**. The Agriculture Victoria Biosciences Advanced Scientific Computing platform (BASC) was used here. It is a Beowulf style cluster that consists of 129 Nodes, with a total of 5832 cores, 103TiB of RAM and 1.5PiB of local scratch disk. 2 × DDN GS14Ks, running Spectrum Scale, that provided 8PiB capable of delivering an aggregate of 50 GB/s. A Full fat-tree network running at 100 Gb/s backbone and 25 Gb/s to each compute node. System software included: OS: Centos Linux 7.6.1810, Resource Manager: SLURM 19.05.2, Software Build System: EasyBuild 4.1.1, and Filesystem: IBM Spectrum Scale 5.0.3–3. Program

were run on a dual socket system with 2 × Xeon Platinum 8168 CPUs, with a total of 48 cores and Hyperthreading disabled. 24 × DDR4-2666 ECC REG DIMM, 24 TB of local scratch space.

**Reporting summary**. Further information on research design is available in the Nature Research Reporting Summary linked to this article.

## Data availability

Requests for data sharing will only be considered for research purposes because the data is owned by Australian dairy farmers. It was obtained through DataGene, https://datagene.com.au/, an Organisation that is responsible for genetic evaluation of dairy cattle in Australia Data used to generate the manuscript Figures are given in Supplementary Data 2.xlsx.

## Code availability

An R code version of BayesR3 is given in the Supplementary Software 1.zip together with example genotype and phenotype data. An executable with usage and example data will be made freely available for research purposes on request.

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

## Acknowledgements

This work is part of a project within the "DairyBio" programme and was funded by Agriculture Victoria (Melbourne, Australia), Dairy Australia (Melbourne, Australia) and the Gardiner Foundation (Melbourne, Australia). We thank farmers for data recording and DataGene staff (Drs Gert Nieuwhof and Kon Konstantinov, Erica Jewell, and Paul Koh) for data access and processing. We thank Dr Bolormaa Sunduimijid for imputation of genotypes. We are also grateful to Dr Ruidong Xiang for testing earlier versions of the BayesR3 software.

## Author contributions

E.J.B. and M.E.G. developed the code and wrote the initial draft of the paper. M.E.G. developed the equations. E.J.B. wrote and developed the executables for BayesR3 software and helped with its development. H.D.D. helped in writing paper, the development of the software, and the analysis. I.M.M. generated the simulated data and undertook quality control of the real data. I.M.M. and M.H.M. performed the analyses on the milk trait data and the comparative analysis with alternative software. P.N.H. and J.E.P. contributed the data and to the analysis of the MIR data set. C.D.T. was instrumental in the deployment of the BayesR3 program. All authors helped in the preparation of the manuscript.

## Competing interests

The authors declare no competing interests.
