## [Peer Review File · Communications Biology]

Reviewers' comments:

Reviewer #1 (Remarks to the Author):

I thought it was an approximation to BayesR at the beginning, but I prefer to consider it as an "exact" method after reading the methods section. I hope the authors can point out where the sped-up come from clearly by making a one-by-one comparison to the original BayesR. e.g., time complexity, for each step. In addition, please also try to explain the reason why the error bar varies for different methods in Figure 1. e.g, much smaller error for BR in Figure (2000, 1A), as well as show the performance with more iterations.

Line 49: remove "they"

Line 57-59: It is a little bit weird to present the comparison between previous methods and your proposed method here, because the proposed method has not been mentioned yet so far. Line 68: Describe what BayesCpi is.

Line 65,69: The new RHS of what. It may be better to talk about RHS in the method part or further explain it in the introduction part.

Line 72 "RHS-updating", again RHS is not described. I would suggest further improvement on this paragraph.

Line 77 ", " should be "."

In both the Method and Result parts, I recommend bold the letters when they represent matrices or vectors.

Line 93: "an" ...

Line 98: it is named as "polygenic variance" in the Method section. Line 106 "additive" genetic variance

Line 128-130 I prefer n_R , n_M .

Line 160 Remove "Figure 1" at the end of the sentence. Line 163 assess?

Figure 1 The label for the y axis label is better to be h^2 .

Line 224 Explain more on the square root of the number of observations is recommended.

Line 277, 669: More explanation is needed for this multi-trait method. In some sense, this may not be biologically meaningful.

Cheng, H., Kizilkaya, K., Zeng, J., Garrick, D. & Fernando, R. Genomic Prediction from Multiple-Trait Bayesian Regression Methods Using Mixture Priors. *Genetics* 209, genetics.300650.2018-103 (2018).

Line 547 "calving"

Line 569 Explain σ_{j}^2 , σ_{t}^2 ,

Reviewer #2 (Remarks to the

Author): Please see attached

file.

TITLE: Bayes R3: fast MCMC blocked processing for large scale multi-trait genomic prediction and QTN mapping analysis

AUTHORS: Breen et al.

The authors use simulated and real cattle data and address estimation of SNP effects and prediction of breeding values (polygenic scores in human applications). The paper claims to present an extension of a mixture model to multiple-traits and considers a block sampling strategy that is seemingly computationally advantageous over more "classical" forms of running a Gibbs sampler for this type of problem. The study reflects a significant amount of work, but is it far from transparent or convincing.

First, it is unclear why they claim their approach is a multiple-trait one, when the assumption "must be" (see text) that traits are uncorrelated! A real multiple-trait involves specifying covariance matrices (across traits) for each random effect fitted in the model. No such matrices arise at all in the presentations of methods and of results. Actually, they just report estimates of single trait parameters, so evidence is lacking that the approach works with mundane multivariate plant breeding data, say.

Second, the model description is not described with enough detail. It would be hard for many readers to understand what is going on without digging considerably in the literature or from earlier papers on this sequence of 4-component mixtures entitled Bayes R, Bayes3, Bayes3+, BayesRC, which has seemingly been advanced by the research group in the last decade. I got confused trying to make sense of what is what, as the authors did not help much in terms of sign posting. I would be incapable of writing my own code to reproduce what the authors propose, based on the lack of detail in the paper. I would be at a loss if I actually had to do real-world multivariate predictions.

There is no evidence presented that supports why a 4-component with the sequence of variances mentioned is needed. The authors do not suggest or provide any tests for arriving at the proposed model. It sounds as if the genome can be universally represented (mathematically) as a 4-component mixture. I do not see any obvious reason of why the variance parameters are

set at the values they use, as opposed to others. No argument or evidence is presented that one should set variances at 0, 10^{-4} , 10^{-2} and so on. Perhaps a 2-component model would receive strong data support or even display a better predictive performance. Perhaps the variances should be different... Perhaps the spike and slab may involve distributions other than the normal... Perhaps none of the above.

Their proposed methodology is compared against BLUP, which induces a homogeneous shrinkage of effects. However, mixture models are not the only way of dealing with a large collection of heterogeneous effects, mostly tiny (arguably not zero) but some large. There are multivariate competitors published in the literature that may produce a more interesting (and perhaps relevant) comparison. See below.

MCMC is a powerful algorithm (not an estimation method) and it is important to test for convergence. The only test conducted in the study seems to be "looking at" a trace plot and, once it stabilizes, the run is considered OK by the authors. This is naïve, scientifically insufficient and dangerous. Further, there is software for tests for convergence and publicly available. Computer speed is important but there is no point in getting fast to a wrong destination! I am not saying that the results are wrong. Rather, the impression is that there is no satisfactory evidence that the draws are truly from the posterior distribution. It is conceivable that the estimates of breeding value are OK but estimates of SNP effects may not be so especially for the "null or low" variance classes. One should expect a better mixing for linear combinations of effects than for single-site effect (which are not statistically estimable, anyhow, whenever $n < p$).

The paper is compact and contains a number of unclear sentences and typos. The reader is flooded with analyses and acronyms which are hard to digest. Also (personal preference) it is hard to appreciate the paper implications without first understanding the methods, which come at the end of the paper, after the readers have been bombarded with graphs and tables that must be read in a non-contextual manner. In other words, the results cannot be understood without previously understanding the methods used. What is the point of reading a paper backwards? Even if the paper were reorga-

nized, the entire contribution would lack clarity because the methodological description is sketchy and not rigorous enough.

Some miscellaneous comments follow.

23-24 "increase" relative to what?

32 "putative causal variants". Establishing causality requires further analysis.

35 "treated as random variables"

6 Refer to Habier et al. (2011, BMC Bioinformatics), an important reference to spike-slab distributions in genomic selection.

48 "Bayes R MAY provide".

54 MCMC is not an estimation method. It is a sampling ALGORITHM that allows estimation of aspects of a posterior distribution from the Monte Carlo sampling. A common estimator of the posterior expectation is the mean of the post-burn in samples.

90 This is a UNIVARIATE model but the title claims "multiple-trait". In fact (refer to the methods), there is no evidence that their analyses are handling multi-response data. Actually, they form PCA "phenotypes" or deal with uncorrelated variables. There is no evidence in the manuscript that there is handling of either environmental or genetic (across-trait) covariance matrices, which is the crux of a multivariate approach. I maybe missing something, but neither the title nor the claims of a multi-trait analysis seem justified.

93 Often the human and animal data are observational and not from a randomized trial. Replace "design" by "incidence".

94 It would be sensible to point that a is a vector of effects with same covariance matrix (otherwise, it is not separable from the residual). You state that in the "Methods" section, but it should be pointed out here. Again, note that for a TRULY multivariate approach your variance parameters should be replaced by covariance matrices. I failed to see this in the paper, throughout.

104 The authors adopt a 4-component mixture model with arbitrary variances without ANY justification. Have they tested for 2, 3 or more than 4 components? Is there a biological rationale for this representation? In

statistics (and science), model parsimony is important.

111 Vg seems to suggest "genetic variance" which is obviously not the case. It would help readers if authors followed the classical notation of boldfacing matrices (upper case) and vectors (lower case).

119 I do not think that a test of "predicted SNP effects" can be derived from "simulated true breeding values". Actually, in their setting ($\#RECORDS < \#SNPs$) the individual SNP effects are not estimable and the "estimates" cannot be disentangled from the prior (e.g., Gianola 2013, Genetics). The statement is misleading.

128-130 Mentioning that $n = 25$ twice is redundant and confusing. Re-write.

149-152 The authors expect that a slope equal to 1 is indicative of "no bias". This would be so in a classical frequentist setting: OVER AN INFINITE NUMBER of samples of the process, the expected value of predictions must be equal to the expected value of the true values, should the model hold. However, the simulations and real data analysis in this paper are based on a single realization of either SNP effects or breeding values, so bias is to be expected most of the time. Actually, a conditional expectation is said to be unbiased if its average value (over an infinite number of replications) is equal to the average (over infinite replications) predictand; this is shown in many papers by Harville, Searle and Henderson. An estimator (of a fixed parameter cp) is unbiased if for some estimator cp , its expected value is $E(cp) = cp$. For a predictor u of a RANDOM QUANTITY, the definition of unbiasedness is $E(u) = E(u)$. It is easy to show that $E(u) \neq u$ (thus, bias with respect to u), and that $E[E(u)] = E(u)$ where the outer iterated expectation is taken with respect to the data distribution. This latter expectation does not arise in the Bayes treatment of problems, as all inferences are based on the available data, not on imaginary data (violation of the likelihood principle). In summary, it would not be surprising if, in an analysis or a single simulation or data set, the slope differs from 1. It should, unless each of the elements of u has a large information content (e.g., a clones with large numbers of replicate measures). I urge the authors to be as precise as possible and avoid hand-waving arguments when these are

unneded.

158-160 It may well reflect the sample, but in finite samples, neither REML nor posterior means are unbiased. They are consistent and asymptotically unbiased. The authors are implicitly stating that these methods are unbiased: they are not.

163 "assess" not "access"

174-180 These statements do not reflect what is shown in Figure 3. You state that the comparison is between estimated and true effects, but this is not shown at all. Actually, Figure 3 gives effect sizes vs genomic positions and the correlations presented pertain to estimated breeding values and true breeding values. I would expect the correlation between true and estimated SNP effects to be much lower across ALL SNPs, than the 0.56-0.76 given in the figure. If one cherry picks SNP effects with non-zero effects and their estimates, the correlations will be higher than across all SNPs but perhaps lower than the correlations between breeding values and their estimates. In short, the discussion and the Figure do not inform about the extent to which the SNP effects are suitably recovered by the model. A more informative figure could present an association smoothed scatter diagram showing the extent of alignment between true values and their estimates separately for signal and non-signal effects.

179-180 It appears to me that, for complex traits, it is also relevant to ask on the extent to which the small effect variants can be recovered, if at all. Your mixture model may severely mitigate small effect variants! Curiously, your parameterization of the mixture is done by allocating different variances to the four classes, but the same mean. If you have a SNP with a very large effect but with a small variance (i.e. in class 1 a priori) where would it end a posteriori?

183-185 Where are these correlations presented?

Figure 1 Can you please help the reader and explain what BayesR, BR3 and BR3+ are? I do not see a clear description of the differences between the 3 methods. Presumably all are mixture models with different structures. Help needed here.

Figure 1 Are these "accuracies" correlations in a training set (i.e., goodness

of fit) or from some cross-validation? Also it appears that σ^2 is fixed externally, i.e., not estimated from the data at hand. Is it reasonable to assume σ^2 known? Do you describe how this parameter is estimated?

250-252 and Table 2. Please try to explain this better and either discuss proportions or numbers. If you proportions are discussed, the Table should present proportions! Incidentally, why are there some non-integer numbers in Class 2 of table 2?

265-271 Is this real data (I assume it is) but there is so much going on in the paper that is hard to keep track. Sing post please!

267 Delete "equally".

276-277 I fail to see the point of illustrating a multivariate analysis with uncorrelated traits. It is clearly inconsistent with the title of the paper.

281 Can you explain this more clearly? Where do these "two additional variables" appear? Throughout the paper, the description of how the mixture model is implemented lacks transparency.

282-286 Cannot follow this explanation.

295 Figure 6d is barely legible.

295-296 What are you trying to say here? If some SNPs affect several traits, there should be correlation. The results are contradicting the orthogonality of PCs?

369-370 The sentence is nonsensical. There is no such thing "as prior knowledge of a posterior probability". Do you mean that a posterior probability distribution (based on some data) was used as prior for a different data set?

398 Replace the strong statement "can describe the genetic architecture" by "can help to understand the genetic basis". Avoid grandiose claims.

398-410 Here, and apparently in previous papers, the authors settle without much justification on a 4-component model with classes that have a zero mean and increasing "prior variances". Why 4 groups? Why the four arbitrarily chosen variances? There should be an attempt to give some rationale for this choice. Perhaps 2 or 3 classes are better...Also, why not let the data drive the number of classes? Dirichlet process priors may be an

interesting alternative approach (e.g., Van der Merwe and Pretorius, GSE 35: 137-158. Some discussion of this approach may be interesting.

404-405 Is it not entirely fair to compare a model that makes allowance for clusters in the distribution (e.g., Bayes R) against one that uses a single cluster (e.g., GBLUP). In the latter, the variants with strong effects are excessively tempered. I suspect that a two-stage approach in GBLUP, 1) select significant variants with any simple model and 2) fit them as fixed effects in GBLUP may give similar results (prediction at least) as the mixture model presented here (I may be wrong but it would be nice to know). Also, there are alternative methods such as the multiple-trait mixture model of Cheng et al. (2018, GENETICS 209 89-103) and the multivariate Lasso in Gianola and Fernando (Genetics 214: 305-331) which do SNP selection and differential shrinkage and these are ignored in the discussion. As a minimum, these papers need to be cited, to make the literature review a little less parochial.

424 Not sure if you have defined R .

428 "thread" of what? Also what do you mean by "original method"?

434 Do not understand: "they are largely explained the same SNP"

?? Again, what is the point of a multiple-trait analysis when traits are uncorrelated? A true multiple-trait analysis involves structured covariance matrices (e.g., Cheng et al. 2018)

443 Replace "there is a causal" by "there may be a causal variant".

445 I do not think this sum is "legal". Recall $P(A \cup B) = P(A) + P(B) - P(A \cap B)$. More generally,

$$P(A \cup \dots \cup K) = \sum_{i=1}^K P(A_i) - \sum_{i < j=2}^K P(A_i \cap A_j) + \sum_{i < j < k=3}^K P(A_i \cap A_j \cap A_k) + \dots + (-1)^{K-1} P(A_1 \cap A_2 \cap \dots \cap A_K).$$

Since a SNP can enter either alone or with other distinct SNPs at any iteration, it is unclear what the sum of probabilities measures, as it ignores joint events.

449-451 I am not sure this was explained with enough clarity in the

methodology.

459-461 Another possibility is that the mass point of SNPs at zero with no variance may be an influential part of the model than essentially nullifies small-size variants whenever $n < p$. Perhaps there is a deeper theoretical investigation of this type of problem in the statistical literature. Notsure.

474 Is this statement valid for situations when a large number of individuals are not genotyped? In other words, are you claiming that your methodology would have the same computing performance as a "single-step" (Aguilar-Misztal-Legarra) multi-trait analysis?

500-501 This paragraph confirms that the simulation does not pertain to a multiple-trait setting as no environmental or genetic correlations were introduced in the data generating process.

543 First derivatives of what?

553 Replace "normalised" by "scaled and centred". If v_{ij}^* is discrete, the distribution of v_{ij} is discrete as well, so it cannot be normal!

558 Replace "solve" by "implemented". Models are not solved, equations are!

560 Replace "design" by "incidence"

565 If the assumption is $a \sim N(0, \sigma^2)$ please state it clearly. Formulae after 569. Do you take σ^2 and h^2 as known? Is this an influential assumption?

572-573 As far as mathematical statistics goes, there is no such thing as a scaled inverse chi-square distribution with negative degrees of freedom. I think your priors for the variances are uniform.

579-585 This is a rather unclear (perhaps too compact) description. Presumably $7r$ is vector containing $4c$ counts where c is the number of traits? What is the joint prior? Why do you describe how you sample this vector without giving a clear description of the model structure for a multiple-trait setting?

591-593 Normally, phenotypic variances and covariances are estimated from a model with a structure. Here you ignore the hierarchical structure (i.e., the random effects) and propose a contrived estimator of the

"raw" variance. I am confused.

618 "Least-squares"? You mean that it has the form of a least-squares estimator?

637 What is the justification for fitting a G matrix when SNPs are already accounted for in the mixture model specification. This is unusual and should not be introduced without

explanation.

669-676 Here we encounter again the stumbling block mentioned earlier in the review. You claim a multi-trait model but the traits must be assumed uncorrelated!!!! Also, what is a Cholesky trait? The Cholesky transformation is well known, but I do not know what you mean by a "Choleskytrait"???

Expressions following 683. Where is all this coming from? It is dropped like a bomb from the sky without any explanation whatsoever.

- 688 What is a "proportional probability"?
- 691 What is an "uninformed" prior?
- 694 Where is the multinomial distribution?
- 643 Where do 0.1 and 0.9 come from?

Reviewer #3 (Remarks to the Author):

Dear authors,

I have finished reading the manuscript and I think it is very interesting.

I have several comments, that can help you to improve the manuscripts, that you can find below.

COMMENTS

Line 45-53, There exists other models that assign mixture distribution to marker effects, for example BayesC, BayesB, which is the advantage of BayesR?.

Lines 104, 105. How robust is the method when setting different values for the constants that multiply σ_g^2 , i.e., 10^{-4} , 10^{-3} , 10^{-2} ?

Line 108, why these hyperparameters?. Please explain or add a reference.

Lines 109, 128-130 How to select the number of blocks and the number of markers included in each one?.

Figure 2, move the legend so that we can see the bars. Perhaps a more informative plot can be built based on estimated n 's for each of the classes. Compare the true n 's with the estimated ones.

Figure 3, Add an extra figure where you plot the true SNP effect vs Estimated SNP effect.

Figure 5, Remove the titles from Figure 5 and use the letters a, b to give the description in the legend.

Line 564, "Z is an identity matrix", well if the phenotypical records and the rows/columns of matrix A are properly sorted, yes, but is not always the case. Also in other contexts (e.g. plant breeding) you can have replicates, so Z not necessarily is an identity matrix.

Lines 569-570, replace "*" with the times operator so that you have the same notation in all the document.

Lines 570-571: Describe σ_t^2 . Which is the prior assigned to σ_g^2 ?

Lines 572-573, the use on improper priors can cause convergency issues, comment about this, and give reference to justify the selection of hyperparameters. Furthermore, there are two parameterizations for scaled inverse chi-squared distributions (e.g. Sorensen & Gianola, 2002). Which is the one used here?

Matrix W has not been described... in the weighted analysis you describe a matrix of weights, I assume that in the case of univariate analysis W is the identity matrix

References:

Sorensen, Daniel, and Daniel Gianola. 2002. Likelihood, Bayesian and MCMC Methods in Quantitative Genetics. Statistics for Biology and Health. New York: Springer-Verlag.

Standardize the notation in mathematical expressions, sometimes the matrixes/vectors appear with boldface font, and in other cases without boldface, for example equations in lines 569-573.

How do you check convergence to the stationary distributions for BayesR3?

Can you add a section with 5 cross-validation analysis to show how well BayesR3 predicts?. You can compare it with GBLUP for example.

Line 736: Code availability

The code is not available, what is available is the software.

Nowadays it is a widely used practice to provide the source code that implements proposed models, so that others can adapt the programs for other problems. Are you planning to upload the code and make it available to the public using one of the existing licenses (e.g. GNU-GPL, BSD, etc.)?. In the other hand, if you just provide the executables, it will be difficult to run that executable because very often linux distributions do not include the same library versions and are not always installed in the same directories, so one executable that was generated using Linux Centos, will not run on Linux Ubuntu, etc.

My suggestion is that you upload the code to a public site, e.g. github.

Response to Reviewer 1:

Dear Reviewer, 1

We would like to thank you for all the time that you obviously spent in the review process on our manuscript. We are most gratefully for all the useful and constructive comments, all of which I believe will go towards strengthening the scientific soundness and improves the communication quality of our work.

With respect to the new manuscript, new text to that of the original document is highlighted in blue while line numbers with such changed text are highlighted with a vertical bar.

Below we give our response, and please note that your questions and comments are in bold font, while our replies are not.

Regards

Edmond Breen.

Reviewer #1 (Remarks to the Author):

I thought it was an approximation to BayesR at the beginning, but I prefer to consider it as an "exact" method after reading the methods section. I hope the authors can point out where the sped-up come from clearly by making a one-by-one comparison to the original BayesR. e.g., time complexity, for each step. In addition, please also try to explain the reason why the error bar varies for different methods in Figure 1. e.g, much smaller error for BR in Figure (2000, 1A), as well as show the performance with more iterations.

With respect to Figure 1a variances for BR being smaller than that seen for BR3 by iteration 2000, this we believe is data set dependent as Figure 1b doesn't show this behaviour.

With respect to speed analysis, Figure 5a does provide a head-to-head comparison between BR and BR3. However, to make things clearer we have added the following text to the section for the "Effect of Block Size on Processing Speed" (line 258):

Therefore, for a given number of records we conclude that the processing time for BayesR3 (BR3) is proportional to $\left(\frac{n_R+n}{n}\right)$ per SNP, while for BayesR (BR) the time required is expected to be proportional to n_R per SNP.

See lines 265 to 267.

We have also address in greater detail the expected time complexity of BR3 in a newly supplied supplementary document titled: "Optimizing block size and number of inner cycles.pdf". In brief, the increased speed of Bayes R3 derives from performing several samples of SNP effects for one processing of the phenotypic records.

Line 49: remove "they"

Changed:

genetic architecture of complex trait, **they** map the causal ...

To:

genetic architecture of complex trait, map the causal ...

See line 49

Line 57-59: It is a little bit weird to present the comparison between previous methods and your proposed method here, because the proposed method has not been mentioned yet so far.

Changed:

A recent approach (18) uses parallelization to sample SNP effects. This approach requires many computer cores, to approach the speed of the single core approach as presented here and requires nearly twice the RAM.

To

A recent approach (19) advocated estimating SNP effects in parallel, but it required many computer nodes, hundreds of cores and twice the expected memory (RAM), thereby restricting its utility to all but the largest computer facilities.

See lines 57-60

Line 68: Describe what BayesCpi is.

We removed the reference to BayesCpi as it didn't add to the introduction.

Line 65,69: The new RHS of what. It may be better to talk about RHS in the method part or further explain it in the introduction part.

We modified the introduction sentence:

Calus (19) sped up the Bayesian MCMC procedure SSVS (stochastic search variable selection) by processing about 5 or 6 SNPs at a time so that updating of residuals and computing new right hand sides was only done every 5 or 6 SNPs.

To:

Calus (20) sped up the Bayesian MCMC procedure SSVS (stochastic search variable selection) by processing 5 or 6 SNPs at a time.

See line 65

Line 72 “RHS-updating”, again RHS is not described. I would suggest further improvement on this paragraph.

Changed:

Still, MCMC using **RHS-updating** remains slower than a BLUP solution (24).

To

Still, these MCMC updating procedures remain slower than a BLUP solution (24).

See Line 70

Line 77 “,” should be “.”

The line:

information on the SNP effects, **we** demonstrate ...

Now reads:

information on the SNP effects. We demonstrate ...

see line 76

In both the Method and Result parts, I recommend bold the letters when they represent matrices or vectors.

This has not been done as nearly all equations are in matrix and or vector form and this would mean every equation would be in bold font. I fail to see how that makes any difference in the equation interoperability. In mathematics the only fixed tradition is that you say what a variable is. If it's a matrix you say that, if it's a scalar you say that etc. Representing different data types in italics, bold font, different fonts, different alphabets, and symbols is at the discretion author.

Line 93: “an” ...

If the review is suggesting that the beginning ‘a’ on line 93 (now line 90) should be replaced by an ‘an’ then we aren't sure about this, no online grammar checker has suggested this change. Nor does the grammar checker in word flag this as a problem either. Our understanding is that “an” is placed in front of words that begin with a vowel, or a vowel sound.

Line 90:

a ($n_R \times n_F$) incidence matrix, u is a $n_F \times 1$ vector of fixed effects and n_F is the number of fixed effects ...

Line 98: it is named as "polygenic variance" in the Method section.

Changed:

not explained by the SNPs with **additive variance** represented as σ_a^2

To

not explained by the SNPs with **polygenic variance** represented as σ_a^2

See line 94

Line 106 “additive” genetic variance

Changed:

Where σ_g^2 and is the genetic variance explained by the SNPs cumulatively.

To:

Where σ_g^2 is the **additive** genetic variance explained by the SNPs cumulatively.

See line 103.

Line 128-130 I prefer n_R, n_M.

We do use n_R and n_M for the number of records and number of markers (SNPs) respectively. We have also dropped the use of *bs* for block size and replaced it with *n* and we use *m* for the number of outer cycles.

Line 160 Remove “Figure 1” at the end of the sentence.

Changed:

produced for H10 and H30: 0.129 ($\sigma = 0.011$) and 0.32 ($\sigma = 0.013$) respectively, **Figure 1**.

To

produced for H10 and H30: 0.129 ($\sigma = 0.011$) and 0.32 ($\sigma = 0.013$) respectively.

See line 198

Line 163 assess?

Changed:

mixture of normal distributions so we can **access** the ability of BayesR3 to recover this

To:

mixture of normal distributions so we can assess the ability of BayesR3 to recover this

See line 201

Figure 1 The label for the y axis label is better to be h^2.

Figure 1 now is:

See line 129

Line 224 Explain more on the square root of the number of observations is recommended.

We have added a supplementary file: "optimizing block size and the number of inner cycles.pdf" which gives two proofs for this recommendation.

Therefore, we suggest and have determined that block size and the number of inner iterations to be equal and be no greater than the square root of the number of records (see supplementary file: "optimizing block size and the number of inner cycles").

See lines 272-275.

Line 277, 669: More explanation is needed for this multi-trait method. In some sense, this may not be biologically meaningful.

Changed:

The multi-trait model for BayesR, as presented by Kemper et al (25), specifies an indicator variable J . When $J = 0$, the current SNP under consideration has zero effect on all traits. However, if $J = 1$ then the SNP can be associated with one or more traits, and in this case the SNP effect is sampled for each trait independently. It is assumed that the traits are uncorrelated or decorrelated, by being replaced by their respective PCA components, or

Cholesky traits (25, 46). In BayesR, when a SNP is processed, first J is sampled based on all traits and if $J = 1$ the effect on each trait is sampled independently as in the single trait case.

To:

The multi-trait model for BayesR, was presented by Kemper *et al* (25) but we will describe it briefly for completeness. The model for each trait is the same as described for the single trait case. That is using subscript t to indicate trait $y_t = Xu_t + Vg_t + Za_t + e_t$. It is assumed that the traits are uncorrelated or decorrelated, by being replaced by their respective PCA components, or Cholesky transformed traits (25, 47). However, the multi-trait model includes an indicator variable J . When $J = 0$, the current SNP under consideration has zero effect on all traits. However, if $J = 1$ then the SNP can be associated with one or more traits, and in this case the SNP effect is sampled for each trait independently. Therefore, there is a vector π for each trait which stores the probabilities that $k = 1, 2, 3$ or 4 for that trait. However, the interpretation of π is slightly different to the single trait case because it is the probability conditional on $J = 1$. In BayesR, when a SNP is processed, first J is sampled based on all traits and if $J = 1$ the effect on each trait is sampled independently as in the single trait case.

See lines 732-744

Cheng, H., Kizilkaya, K., Zeng, J., Garrick, D. & Fernando, R. Genomic Prediction from Multiple-Trait Bayesian Regression Methods Using Mixture Priors. *Genetics* 209, genetics.300650.2018-103 (2018).

Our method does not assume that a SNP affects all traits or none of them. If a SNP is sampled into the model on a given cycle its effect is sampled for each trait, and this includes the possibility that it has no effect on one or more traits.

Line 547 “calving”

Changed:

.. and age of **caving**, ...

To

... and age of calving, ...

See line 595

Line 569 Explain $\sigma_{t_j}^2$, $\sigma_{t_j}^2$,

We have added the follow text to the manuscript

The trait variance σ_t^2 is the estimated trait variance obtained from the data, and its exact form is given below (see Material and Methods: Weighted Analysis). Note, σ_j^2 is the SNP variance which is sampled from the k^{th} mixture distribution that the j^{th} SNP is in.

See lines 630-632

We have also changed:

$$\sigma_t^2 = \frac{\sum_{i=1}^{n_t} \left(\frac{(y_{i,t} - \bar{x}_t)^2}{h_t^2 + \frac{1-h_t^2}{w_{i,t}}} \right)}{n_t - 1}, \text{ which resolves to the raw phenotypic variance when } w = 1.$$

To

$$\sigma_t^2 = \frac{\sum \left(\frac{(y_{i,t} - \bar{x}_t)^2}{h_t^2 + \frac{1-h_t^2}{w_{i,t}}} \right)}{n_W - 1}; \quad \forall i, w_{i,t} \neq 0. \text{ Note } n_W \text{ is the number of non-zero weights, and } \sigma_t^2 \text{ resolves to the raw phenotypic variance when } w = 1.$$

See Line 644

Response to Reviewer 2:

Dear Reviewer, 2

We would like to thank you for all the time that you obviously spent in the review process on our manuscript. We are most gratefully for all the useful and constructive comments, all of which I believe will go towards strengthening the scientific soundness and improves the communication quality of our work.

With respect to the new manuscript, new text to that of the original document is highlighted in blue while line numbers with such changed text are highlighted with a vertical bar.

Below we give our responses to your comments, and please note that your questions and comments are in bold font, while our replies are not.

Regards

Edmond Breen.

Reviewer #2 (Remarks to the Author):

TITLE: Bayes R3: fast MCMC blocked processing for large scale multi-trait genomic prediction and QTN mapping analysis

AUTHORS: Breen et al.

The authors use simulated and real cattle data and address estimation of SNP effects and prediction of breeding values (polygenic scores in human applications). The paper claims to present an extension of a mixture model to multiple-traits and considers a block sampling strategy that is seemingly computationally advantageous over more "classical" forms of running a Gibbs sampler for this type of problem. The study reflects a significant amount of work, but is it far from transparent or convincing.

First, it is unclear why they claim their approach is a multiple-trait one, when the assumption "must be" (see text) that traits are uncorrelated! A real multiple-trait involves specifying covariance matrices (across traits) for each random effect fitted in the model. No such matrices arise at all in the presentations of methods and of results. Actually, they just report estimates of single trait parameters, so evidence is lacking that the approach works with mundane multivariate plant breeding data, say.

Second, the model description is not described with enough detail. It would be hard for many readers to understand what is going on without digging considerably in the literature or from earlier papers on this sequence of 4-component mixtures entitled Bayes R, Bayes3, Bayes3+, BayesRC, which has seemingly been advanced by the research group in the last decade. I got confused trying to make sense of what is what, as the authors did not help much in terms of sign

posting. I would be incapable of writing my own code to reproduce what the authors propose, based on the lack of detail in the paper. I would be at a loss if I actually had to do real-world multivariate predictions.

There is no evidence presented that supports why a 4-component with the sequence of variances mentioned is needed. The authors do not suggest or provide any tests for arriving at the proposed model. It sounds as if the genome can be universally represented (mathematically) as a 4-component mixture. I do not see any obvious reason of why the variance parameters are set at the values they use, as opposed to others. No argument or evidence is presented that one should set variances at 0; 10^{-4} ; 10^{-2} and so on. Perhaps a 2-component model would receive strong data support or even display a better predictive performance. Perhaps the variances should be different... Perhaps the spike and slab may involve distributions other than the normal... Perhaps none of the above.

Their proposed methodology is compared against BLUP, which induces an homogeneous shrinkage of effects. However, mixture models are not the only way of dealing with a large collection of heterogeneous effects, mostly tiny (arguably not zero) but some large. There are multivariate competitors published in the literature that may produce a more interesting (and perhaps relevant) comparison. See below.

MCMC is a powerful algorithm (not an estimation method) and it is important to test for convergence. The only test conducted in the study seems to be "looking at" a trace plot and, once it stabilizes, the run is considered OK by the authors. This is naïve, scientifically insufficient and dangerous. Further, there is software for tests for convergence and publicly available. Computer speed is important but there is no point in getting fast to a wrong destination! I am not saying that the results are wrong. Rather, the impression is that there is no satisfactory evidence that the draws are truly from the posterior distribution. It is conceivable that the estimates of breeding value are OK but estimates of SNP effects may not be so especially for the "null or low" variance classes. One should expect a better mixing for linear combinations of effects than for single-site effect (which are not statistically estimable, anyhow, whenever $n < p$).

The paper is compact and contains a number of unclear sentences and typos. The reader is flooded with analyses and acronyms which are hard to digest. Also (personal preference) it is hard to appreciate the paper implications without first understanding the methods, which come at the end of the paper, after the readers have been bombarded with graphs and tables that must be read in a non-contextual manner. In other words, the results cannot be understood without previously understanding the methods used. What is the point of reading a paper backwards? Even if the paper were reorganized, the entire contribution would lack clarity because the methodological description is sketchy and not rigorous enough.

Reviewer 2 doesn't appreciate that we are just following the Journal's expected format. However, we have made attempts to address reviewer 2's concern in our responses to his individual questions and comments given below:

Some miscellaneous comments follow.

23-24 "increase" relative to what?

Changed:

mid infra-red spectra data as an example of "omics" data and show its use to increase the precision of mapping variants affecting milk, fat, and protein yields.

To:

mid infra-red spectra data as an example of “omics” data and show its use to increase the precision of mapping variants affecting milk, fat, and protein yields relative to a univariate analysis of milk, fat, and protein.

See line 23-25

32 "putative causal variants". Establishing causality requires further analysis.

This is in the introduction section, and we are referring here to the true causal variants and not putative causal variants.

35 "treated as random variables"

Changed:

where the effects on the trait, or phenotype, are random variables

To:

where the effects on the trait, or phenotype, are **treated as random variables**

See line 35

36 Refer to Habier et al. (2011, BMC Bioinformatics), an important reference to spike-slab distributions in genomic selection.

Changed:

This analysis is known as genomic selection or genomic prediction (GP) (6)

To

This analysis is known as genomic selection or genomic prediction (GP) **(6,7)**

See line 36

Added:

7. Habier D, Fernando RL, Kizilkaya K, Garrick DJ. Extension of the bayesian alphabet for genomic selection. BMC Bioinformatics. 2011;12:186.

See line 847

48 "Bayes R MAY provide".

Changed:

Genomic prediction methods such as BayesR, provide information about the genetic

To:

Genomic prediction methods such as BayesR **may provide** information about the genetic

See line 48

54 MCMC is not an estimation method. It is a sampling ALGORITHM that allows estimation of aspects of a posterior distribution from the Monte Carlo sampling. A common estimator of the posterior expectations the mean of the post-burn in samples.

Changed:

SNP effects are commonly estimated by Markov Chain Monte Carlo (MCMC), but MCMC methods are slow.

To:

SNP effects are commonly estimated using a Markov Chain Monte Carlo (MCMC) **algorithm**, but MCMC methods are slow

See line 54

90 This is a UNIVARIATE model but the title claims "multiple-trait". In fact (refer to the methods), there is no evidence that their analyses are handling multi-response data. Actually, they form PCA "phenotypes" or deal with uncorrelated variables. There is no evidence in the manuscript that there is handling of either environmental or genetic (across-trait) co- variance matrices, which is the crux of a multivariate approach. I may be missing something, but neither the title nor the claims of a multi-trait analysis seem justified.

We have modified the introduction section outlining the multi-trait model in the material and methods to read:

The multi-trait model for BayesR, was presented by Kemper *et al* (25) but we will describe it briefly for completeness. The model for each trait is the same as described for the single trait case. That is using subscript t to indicate trait $y_t = Xu_t + Vg_t + Za_t + e_t$. It is assumed that the traits are uncorrelated or decorrelated, by being replaced by their respective PCA components, or Cholesky transformed traits (25, 47). However, the multi-trait model includes an indicator variable J . When $J = 0$, the current SNP under consideration has zero effect on all traits. However, if $J = 1$ then the SNP can be associated with one or more traits, and in this case the SNP effect is sampled for each trait independently. Therefore, there is a vector π for each trait which stores the probabilities that $k = 1, 2, 3$ or 4 for that trait. However, the interpretation of π is slightly different to the single trait case because it is the probability conditional on $J = 1$. In BayesR, when a SNP is processed, first J is sampled based on all traits and if $J = 1$ the effect on each trait is sampled independently as in the single trait case.

See lines 732-744,

Also, reviewer 2 seems to have a singular view about what a multi-trait analysis is. However, our view is that reviewer 2 is incorrect about there being only one way to do a multi-trait analysis. Another way to carry out a multi-trait analysis is to transform the traits so that they are uncorrelated. In general, this can be done with a canonical transformation which makes the transformed traits uncorrelated at both the genetic and residual level. This requires estimates of the residual and genetic covariance matrices. With many traits, sampling errors in the estimated genetic covariance matrix build up until it becomes non-positive definite. Even if the genetic covariance matrix remains positive definite the errors of estimation have serious effects on the estimates of other parameters. Often, the genetic and residual covariance matrices are alike to within a constant of proportionality. Therefore, to avoid using a poorly estimated covariance matrix, we assume this

property. In this case, a transformation that makes the phenotypes uncorrelated will approximately make the transformed traits uncorrelated on both the residual and genetic levels. Thus, our use of a transformation to uncorrelated traits is an approximation for a full multi-trait analysis.

93 Often the human and animal data are observational and not from a randomized trial. Replace “design” by “incidence”.

Changed:

a $(n_R \times n_F)$ design matrix, u is a $n_F \times 1$ vector of fixed effects and n_F is the number of

To:

a $(n_R \times n_F)$ **incidence** matrix, u is a $n_F \times 1$ vector of fixed effects and n_F is the number of

See line 90

94 It would be sensible to point that a is a vector of effects with same covariance matrix (otherwise, it is not separable from the residual). You state that in the “Methods” section, but it should be pointed out here. Again, note that for a TRULY multivariate approach your variance parameters should be replaced by covariance matrices. I failed to see this in the paper, throughout.

Added:

such that $a \sim N(0, A\sigma_a^2)$, and A is the additive relationship matrix

to line 95

104 The authors adopt a 4-component mixture model with arbitrary variances without ANY justification. Have they tested for 2,3 or more than 4 components? Is there a biological rationale for this representation? In statistics (and science), model parsimony is important.

A mixture of 4 normal distributions can approximate a wide range of distributions e.g., a reflected gamma or a t-distribution. The method is not limited to 4 distributions in the mixture (any number, $n \geq 2$, can be used) or to the 10-fold pattern of variances. We have found a 10-fold pattern works well perhaps because it is close enough to allow SNPs to move easily between distributions and because the sum of the 4 distributions creates a relatively smooth long-tailed distribution.

We do not include all this justification in the paper because it is not the main purpose of the paper. The use of this mixture of normal distributions in Bayes R has been published in 2011 and the multi-trait version in 2015 so it does not seem helpful to the reader to justify these features here.

111 V_g seems to suggest "genetic variance" which is obviously not the case. It would help readers if authors followed the classical notation of boldfacing matrices (upper case) and vectors (lower case).

The variances in the manuscript are all represented using the square of sigma notation: σ_x^2 where x is used to denote the variance types. We have not set the equations to bold font, simply because all equations are in matrix and or vector form and this would mean every equation would be in bold font. In Mathematics the only fixed tradition is that you declare what a variable is. If it's a matrix you say that, if it's a scalar you say that etc. Representing different data types in italics, bold font, different fonts, Greek letters etc is at the discretion of the author. Also, please note, there is no guidance given by the Journal on the style of mathematical notation that must be used.

119 I do not think that a test of "predicted SNP effects " can be derived from "simulated true breeding values". Actually, in their setting (#RECORDS<=SNPs) the individual SNP effects are not estimable and the "estimates" cannot be disentangled from the prior (e.g., Gianola 2013, Genetics). The statement is misleading.

It is true that estimated SNP effects depend on the priors chosen. We show using simulation that the estimate SNP effects are correlated with the simulated SNP effects. Please refer to Figure 4.

128-130 Mentioning that n = 25 twice is redundant and confusing. Re-write.

Changed:

block size $bs = n = 25$ SNPs and $m \times n$ samples, where $m = n_L$ (i.e., the same number of outer iterations as BayesR), and to BayesR3, where $m = \frac{n_L}{n}$ and $n = 25$.

To

block size $n = 25$ SNPs and $m \times n$ samples are drawn per parameter, where $m = n_L$ (i.e., the same number of outer iterations as BayesR), and to a version (here called BayesR3), where $m = \frac{n_L}{n}$,

See lines 138-140

149-152 The authors expect that a slope equal to 1 is indicative of "no bias". This would be so in a classical frequentist setting: OVER AN INFINITE NUMBER of samples of the process, the expected value of predictions must be equal to the expected value of the true values, should the model hold. However, the simulations and real data analysis in this paper are based on a single realization of either SNP effects or breeding values, so bias is to be expected most of the time. A conditional expectation is said to be unbiased if its average value (over an infinite number of replications) is equal to the average (over infinite replications) predictand; this is shown in many papers by Harville, Searle and Henderson. An estimator (of a fixed parameter $\hat{\varphi}$) is unbiased if for some estimator $\hat{\varphi}$, its expected value is $E(\hat{\varphi}) = E(\varphi)$. For a predictor \hat{u} of a RANDOM QUANTITY, the definition of unbiasedness is $E(\hat{u}) = E(u)$. It is easy to show that $E(\hat{u}|u) \neq u$ (this, bias with respect to u), and that $E[E(\hat{u}|u)] = E(u)$ where the outer iterated expectation is taken with respect to the data distribution. This latter expectation does not arise in the Bayes treatment of problems, as all inferences are based on the available data, not on imaginary data (violation of the likelihood principle). In summary, it would not be surprising if, in an analysis or a single simulation or data set, the slope differs from 1. It should, unless each of the elements of u has a large information content (e.g., a clones with large numbers of replicate measures). I urge

the authors to be as precise as possible and avoid hand-waving arguments when these are unneeded.

We agree with reviewer in that it is not the classical use of the term bias. However, in the dairy literature the word bias is used instead of the words well calibrated. The slope = 1 means the expectation of u , given \hat{u} is equal to \hat{u} . This well calibrated property is a desired property of estimated random effects. Therefore, we have added the sentence:

Changed:

In Figure 1c and Figure 1d the prediction biases are shown, which is given by regression of the TBV on the SNP predicted EBV and where a coefficient of 1 represents no bias. It is seen in Figure 1c and Figure 1d that the bias is generally within 10% of 1, other than at a low number of iterations.

To

As well as high accuracy, we would like the estimated breeding values to be well calibrated. That is, we would like the regression of true breeding value on estimated breeding value to =1. (This regression is often referred to as "bias" although it is not the classical definition of bias). In **Error! Reference source not found.c** and **Error! Reference source not found.d** these biases are shown. It is seen in **Error! Reference source not found.c** and **Error! Reference source not found.d** that the bias is generally within 10% of 1, other than at a low number of iterations.

See lines 160-164

158-160 It may well reflect the sample, but in finite samples, neither REML nor posterior means are unbiased. They are consistent and asymptotically unbiased. The authors are implicitly stating that these methods are unbiased: they are not.

The test of our method is its ability to estimate the true parameter, BayesR gets the same answer as the well-known REML estimator. We believe, our paper is not the place for a discussion on the bias in REML estimators.

163 "assess" not "access"

Changed:

mixture of normal distributions so we can **access** the ability of BayesR3 to recover this

To:

mixture of normal distributions so we can **assess** the ability of BayesR3 to recover this

See line 199

174-180 These statements do not reflect what is shown in Figure 3. You state that the comparison is between estimated and true effects, but this is not shown at all. Actually, Figure 3 gives effect sizes vs genomic positions and the correlations presented pertain to estimated breeding values and true breeding values. I would expect the correlation between true and estimated SNP effects to be much lower across ALL SNPs, than the 0.56-0.76 given in the figure. If one cherry picks SNP effects with non-zero effects and their estimates, the correlations will be higher than across all

SNPs but perhaps lower than the correlations between breeding values and their estimates. In short, the discussion and the Figure do not inform about the extent to which the SNP effects are suitably recovered by the model. A more informative figure could present an association smoothed scatter diagram showing the extent of alignment between true values and their estimates separately for signal and non-signal effects.

Please note Figure 3 is now Figure 4 in the new manuscript.

We find the reviewer comments a little perplexing as there is no cherry picking going on here. We have added the text:

Error! Reference source not found. provides a visual correlation between the true 4000 effects given in **Error! Reference source not found.**a and their estimated SNP effects for the same 4,000 variants using the simulated training set (H30) embedded within three different genotype densities: 400,000 markers **Error! Reference source not found.**b, 40,000 markers **Error! Reference source not found.**c and only the causal variants **Error! Reference source not found.**d.

See lines 213-216

We have also added to Figure 4b, c, d the actual correlations asked for by the reviewer between the causal SNPs and the complete (signal + non signal SNPs):

Figure 1: The true SNP effects for the 4,000 simulated causal variants and their estimated effects using the H30 training data set, embedded within three different genotype densities. All results are from BR3 using five chains, each of length 2,000 and with a block size of 25. The values above b), c), and d), are all Pearson's correlation values. The first value above each of these figures is the correlation between estimated and true breeding values, the 2nd value is the correlation between the true 4,000 causal SNP effect values to their corresponding estimates, while the 3rd value is the estimated SNP effects correlation across all SNPs within each of the analysis to their simulated true values, b) = 400,000 SNPs, c) = 40,000 SNPs and d) = 4,000 SNPs.

179-180 It appears to me that, for complex traits, it is also relevant to ask on the extent to which the small effect variants can be recovered, if at all. Your mixture model may severely mitigate small effect variants! Curiously, your parameterization of the mixture is done by allocating different variances to the four classes, but the same mean. If you have a SNP with a very large effect but with a small variance (i.e. in class 1 a priori) where would it end a posteriori?

Treating the SNP effects as random effects means they are “regressed back” as they are in BLUP. The mixture model causes the apparently large effect effects to be regressed back less severely than the apparently small effects. However, even the effects sampled to be in the smallest distribution are regressed back by less than they would be in a BLUP analysis. We don’t understand the reviewer’s comment about the mean of each distribution. It is normal to assume random effects are sampled from a distribution with mean =0 because SNP effects can be positive or negative depending on which allele is treated as the base.

183-185 Where are these correlations presented?

Sorry these correlations are now given in Figure 4 (see above) and are presented/seen on the top of each panel given as Figure 4b, c and d.

Figure 1 Can you please help the reader and explain what Bayes R, BR3 and BR3+ are? I do not see a clear description of the differences between the 3 methods. Presumably all are mixture models with different structures. Help needed here.

All 3 are implementing the same model but with differences in the MCMC algorithm. BayesR is the standard BayesR that is, it doesn’t use a block structure. BR3 is the blocked BayesR structure that is the subject of this paper, while BR3+ is the same as BR3 except it is configured to do 25 times more total iterations than BR3 and BR. The essential purpose of BR3+ is to show that the inner cycles influence the solution, since BR3+ has the same number of outer cycles to the total number of iterations done by BR.

Changed:

1) BR (grey bars), represents a standard “non-blocked” BayesR configuration where the number of iterations is the same as the Markov chain lengths generated; **2)** BR3 (blue bars) a blocked BayesR, where $m = n_L/n$ and $n = 25$; therefore, the Markov Chain lengths are the same as for BR; and **3)** BR3+ where: $m = n_L$ and $n = 25$

To:

1) BR (grey bars) represents a standard “non-blocked” BayesR configuration; **2)** BR3 (blue bars) a blocked BayesR, where $m = n_L/n$ and $n = 25$; therefore, the Markov Chain lengths are the same as for BR; and **3)** BR3+ where: $m = n_L$ and $n = 25$. For BR3+ the results are graphed against the number of iterations divided by 25 so that the accuracy etc of BR3+ and BR can be compared when BR3+ has the same number of outer cycles as BR has total iterations.

See lines 144-149

Figure 1 Are these "accuracies" correlations in a training set (i.e., goodness of fit) or from some cross-validation? Also, it appears that σ_g^2 is fixed externally, i.e., not estimated from the data at hand. Is it reasonable to assume σ_g^2 known? Do you describe how this parameter is estimated?

These are accuracies for the prediction of simulated true breeding values for Australian Red cows (RDC) after training using simulated Holsteins and Jerseys which don't share any ancestry with the training set.

In the initial version of BayesR presented by Erbe *et al.* (2012), σ_g^2 was fixed throughout the analysis and determined by $\sigma_g^2 = h_t^2 \sigma_t^2$ as it was assumed h_t^2 was known. Later Moser G, *et al.* (2015) suggested that σ_g^2 be determined from the data using $\sigma_g^2 \sim \frac{n_m g' g}{\chi^2(n_m - 2)}$ as mentioned and used in the paper. Note, that the σ_t^2 is only used to set up the initial value(s) of σ_g^2 , after that each σ_t^2 is not used again.

The sampling for σ_g^2 is explained in section "Sampling the variances":

Sampling the variances: after each Gibbs iteration, and according to a sampling schedule (see below) the variances were sampled using inverse scaled Chi-square distributions via:

$\sigma_e^2 \sim \frac{\sum_i (e_i^2 W_{ii})}{\chi^2(n_w - 2)}$, $\sigma_a^2 \sim \frac{a' G^{-1} a}{\chi^2(n_R - 2)}$ and $\sigma_g^2 \sim \frac{n_m g' g}{\chi^2(n_m - 2)}$. Where $\chi^2(x)$ is a Chi-square distribution with x degrees of freedom; n_w is the number of non-zero weights and n_R is the number of records associated with the specified trait, and n_m is the number of SNPs in the model for this iteration, see also (46). That is, $n_m = \sum_{k=2}^4 \beta_k$, where β_k is the number of SNPs in the k^{th} distribution and the end of each outer cycle. After which, all σ_k^2 are recalculated as given above.

lines 711 to 718

250-252 and Table 2. Please try to explain this better and either discuss proportions or numbers. If you proportions are discussed, the Table should present proportions! Incidentally, why are there some non-integer numbers in Class 2 of table 2?

The non-whole number comes about from the averaging of the number of SNPs falling into these distributions at the end of every iteration during post-burn in. It can also come about from the averaging that occurs across these distributions from different chains.

Changed:

Thus, the analysis does discover the difference between the 2 classes but underestimates it. This occurs because of the LD between SNPs in the 2 classes and the large number of SNPs in class 2.

To:

Thus, the analysis does discover the difference between the 2 classes but underestimates it (all the SNPs in class 2 should have zero effect). This lack of power is because of the LD between SNPs in the 2 classes and the large number of SNPs in class 2.

See lines 288-291

265-271 Is this real data (I assume it is) but there is so much going on in the paper that is hard to keep track. Sing post please!

Yes, this is real data.

Changed:

A comparison of BayesR3 to EM-BayesR (15) and to MTG2/GBLUP (27), using 25,000 animals and with genotypes from 717,463 SNPs shows that BayesR3 is as accurate as EM-BayesR,

To:

A comparison of BayesR3 to EM-BayesR (15) and to MTG2/GBLUP (27), using 25,000 animals **with real phenotype** type data for milk, fat and protein yields and 717,463 SNPs real effects shows that BayesR3 is as accurate as EM-BayesR,

See lines 305-307

267 Delete "equally".

Changed:

animals and with genotypes from 717,463 SNPs show that BayesR3 is **equally** as accurate as

To:

animals with real phenotype type data for milk, fat and protein yields and 717,463 SNP effects shows that BayesR3 is as accurate as

Line 306

276-277 I fail to see the point of illustrating a multivariate analysis with uncorrelated traits. It is clearly inconsistent with the title of the paper.

Please see comment to **90**.

281 Can you explain this more clearly? Where do these "two additional variables" appear? Throughout the paper, the description of how the mixture model is implemented lacks transparency.

There is an indicator variable J and another distribution $\pi^* \sim \text{Dirichlet}(\alpha^*)$. These are explained in the multi-trait section:

Changed:

Multi-trait BayesR3, like multi-trait BayesR (25), has two additional variables

To

Multi-trait BayesR3, like multi-trait BayesR (25), has two additional variables **(as explained in the Material and Methods)**

See line 323

282-286 Cannot follows this explanation.

Changed:

A SNP may be determined to not affect any trait or determined to affect a trait and is then independently assigned to one of the four distributions for each trait. Therefore, it is possible but unlikely for a SNP to be deemed associated with the traits but be assigned to the null distribution for all traits.

To:

A SNP may be determined to be excluded from the model (meaning it affects none of the traits) or included in the model in which case it is then independently assigned to one of the four distributions for each trait. Therefore, it is possible but unlikely for a SNP to be included in the model but be assigned to the null distribution for all traits.

See lines 324-328

295 Figure 6d is barely legible.

Please note Figure 6 is now Figure 7, which now is:

Figure 2: MIR PC trait summary results; **a)** the phenotype variance σ_p^2 for each MIR PC trait plotted against PC number. **b)** the estimated heritability for each PC trait. **c)** the number of SNPs per mixture distribution for each PC trait. **d)** the raw counts for the number SNPs per distribution for each trait. Note the sum of counts for each PC trait is 3,995, which is the number SNPs estimated to be associated to the traits

424 Not sure if you have defined R

Please note R has been replaced by the term n_R in the new manuscript.

Changed:

whereas we use blocks of size equal to approximately \sqrt{R} ,

To:

whereas we use blocks of size equal to approximately $\sqrt{n_R}$, where n_R is the number of records

See line 451

428 "thread" of what? Also what do you mean by "original method"?

To avoid confusion, we have changed:

This reduced the time necessary to process the data on all individuals and appears to be competitive with EM Hybrid, when one thread is used. However, we found the original method offered little speed advantage compared to modern matrix routines, which have built-in vectorization, for processing genotype matrices.

To:

This reduced the time necessary to process the data on all individuals and appears to be competitive with the EM-Hybrid version of BayesR. However, we found that storing 4-6 SNP genotypes as a single number offered little speed advantage compared to modern matrix routines with built-in vectorization.

See lines 464-467

434 Do not understand: "they are largely explained the same SNP" ?? Again, what is the point of a multiple-trait analysis when traits are uncorrelated? A true multiple-trait analysis involves structured covariance matrices (e.g., Cheng et al. 2018)

They are largely explained by the same SNPs, meaning that each MIR trait is mostly associated with the same set of SNPs. This is interesting because the traits analysed are uncorrelated since they are principal components derived from the raw MIR data. Thus, the multi-trait Bayes R leads to a more parsimonious explanation than, for instance, a multi-trait BLUP model in which uncorrelated traits are assumed to be independent. Here although they are uncorrelated, the traits are largely associated with the same set of pleiotropic SNPs.

See line 469

443 Replace "there is a causal" by "there may be a causal variant".

Changed:

the model on >90% of MCMC cycles implying **there is a causal** variant linked to these

To:

the model on >90% of MCMC cycles implying **there may be a causal** variant linked to these

See line 481

445 I do not think this sum is "legal".

We do. For example, on any one MCMC cycle the number of SNPs included in the model is a sample from the posterior distribution of the “number of SNPs independently associated with the trait or traits within this 50kb interval”. Thus, the sum of posterior probabilities over the 50kb interval is an estimate of the number of SNPs independently associated with the trait or traits. If each causal variant is perfectly tagged by one of the SNPs, then this is also an estimate of the number of causal variants within the 50kb interval. If two SNPs predict the genotype of a causal variant better than any one SNP, then the number of SNPs independently associated with the trait overestimates the number of causal variants within the 50kb interval. We expect that most causal variants will be in near perfect LD with at least one of the SNPs.

449-451 I am not sure this was explained with enough clarity in the methodology.

This is explained on lines 633-637, which explains how the class structure are used. The use of classes it also well explained by MacLeod IM, Bowman PJ, Vander Jagt CJ, Haile-Mariam M, Kemper KE, Chamberlain AJ, et al. Exploiting biological priors and sequence variants enhances QTL discovery and genomic prediction of complex traits. BMC genomics. 2016;17:144. This paper is cited twice in the manuscript.

BayesR3 can also include prior biological knowledge about the variants being analysed as done in BayesRC (26). It does this by dividing the variants into classes based on this prior information. Then the mixing proportions that describe the distribution of variant effects is estimated separately for each class. When there is only 1 class the model is that for standard BayesR.

See lines 633-637

459-461 Another possibility is that the mass point of SNPs at zero with no variance may be an influential part of the model than essentially nullifies small-size variants whenever $n < p$: Perhaps there is a deeper theoretical investigation of this type of problem in the statistical literature. Not sure.

We agree and think this is an interesting point to raise. The mass point of SNPs at zero means that SNPs can be excluded from the model. However, the smallest distribution allows for SNP with small effect size.

474 Is this statement valid for situations when a large number of individuals are not genotyped? In other words, are you claiming that your methodology would have the same computing performance as a "single-step" (Aguilar-Misztal-Legarra) multi-trait analysis?

The methods as described assumes all individuals are genotyped. None of the equations presented allow for non-genotyped individuals.

Changed:

The new methodology (BR3) decreases substantially the computer time needed for the analysis so that it is now comparable with a GBLUP analysis.

To:

The new methodology (BR3) decreases substantially the computer time **needed for regressing phenotypes against genotypes such** that it is now comparable with a GBLUP analysis.

See lines 522-524

500-501 This paragraph confirms that the simulation does not pertain to a multiple trait setting as no environmental or genetic correlations were introduced in the data generating process.

A multi-trait analysis can be performed by transforming traits into uncorrelated variables such as PCs. Please see answer to **90**.

543 First derivatives of what?

Changed:

and **taking first derivative**, as commonly recommended for MIR studies

To:

and **taking the first derivative of the MIR signal**, as commonly recommended for MIR studies

See line 591-592

553 Replace "normalised" by "scaled and centred". If v_{ij} is discrete, the distribution of v_{ij} is discrete as well, so it cannot be normal!

The variable v_{ij} isn't discrete, it is only v^*_{ij} , which may be discrete depending on input.

Changed:

The elements of V **normalised**: $v_{ij} = \frac{v^*_{ij} - 2p_j}{\sqrt{2p_j(1-p_j)}}$

To:

The elements of V **were scaled and centred**: $v_{ij} = \frac{v^*_{ij} - 2p_j}{\sqrt{2p_j(1-p_j)}}$

See line 601

558 Replace "solve" by "implemented". Models are not solved, equations are!

Changed:

BayesR and Gibbs sampling was used to **solve** the mixed effects model ...

To:

BayesR and Gibbs sampling was used to **implement** the mixed effects model ...

See line 607

560 Replace "design" by "incidence"

Changed:

X is a $(n_R \times n_F)$ **design** matrix,

To:

X is a $(n_R \times n_F)$ **incidence** matrix,

See line 609

565 If the assumption is $a \sim N(0, A\sigma_a^2)$ please state it clearly.

Changed:

Note, that $e \sim N(0, W^{-1}\sigma_e^2)$.

To:

Note, that $e \sim N(0, W^{-1}\sigma_e^2)$ and that $a \sim N(0, A\sigma_a^2)$.

See line 615

Formulae after 569. Do you take σ_t^2 and h_t^2 as known? Is this an influential assumption?

The variance σ_t^2 and heritability h_t^2 are used to set up the initial value(s) of σ_g^2 , and σ_a^2 after that they're not used again. Note as σ_g^2 is updated from the data, the initial values for σ_t^2 and h_t^2 are not that influential. However, starting off with good approximations may help with speeding up convergences.

572-573 As far as mathematical statistics goes, there is no such thing as a scaled inverse chi-square distribution with negative degrees of freedom. I think your priors for the variances are uniform.

The sentence:

The priors for σ_a^2 and σ_e^2 are scaled inverse chi-squares with **scale 0 and degrees of freedom -2**

has been changed to:

The priors for σ_a^2 and σ_e^2 are scaled inverse chi-squares **with uniform priors (44)**.

See line 629

44. Sorensen D, Gianola D, Gianola D. Likelihood, Bayesian and MCMC Methods in Quantitative Genetics. New York: Springer-Verlag; 2002.

Line 963

579-585 This is a rather unclear (perhaps too compact) description. Presumably π is vector containing 4c counts where c is the number of traits? What is the joint prior? Why do you describe how you sample this vector without giving a clear description of the model structure for a multiple-trait setting?

We have removed this section. However, there is a π vector for each trait as in the single trait case and they are regarded as independent. The connection between the traits is the indicator variable J which specifies if a SNP can influence any of the traits. This model is of course not the only possible multi-trait model, but it has some desirable features. It describes the degree of pleiotropy, that is, do SNPs affect multiple uncorrelated trait? If there is widespread pleiotropy when J=1 the probability that a SNP affects trait t is high. However, if there is little pleiotropy the probability a SNP is in the model (ie p(J=1) may be high but the probability that the SNP affects trait t conditional on J=1 is low. Most other multi-trait models do not have this flexibility.

591-593 Normally, phenotypic variances and covariances are estimated from a model with a structure. Here you ignore the hierarchical structure (i.e., the random effects) and propose a contrived estimator of the "raw" variance. I am confused.

The σ_t^2 is simply the variance of the trait. It is used to create the first estimate of $\sigma_g^2 = h_t^2 \sigma_t^2$ after which σ_g^2 is informed from the data using $\sigma_g^2 \sim \frac{n_m g' g}{\chi^2(n_m - 2)}$ and σ_t^2 is no longer used.

See line 711-714

618 "Least-squares"? You mean that it has the form of a least-squares estimator?

We changed the text:

$$\bar{g}_j = \frac{r_{bj} + (V_b' W V_b)_{jj} g_{bj}^i}{(V_b' W V_b)_{jj}}, \text{ which is the effects' least squares estimate.}$$

To

$$\bar{g}_j = \frac{r_{bj} + (V_b' W V_b)_{jj} g_{bj}^i}{(V_b' W V_b)_{jj}}, \text{ which has the form of a least-squares estimator.}$$

See Line 677

637 What is the justification for fitting a G matrix when SNPs are already accounted for in the mixture model specification. This is unusual and should not be introduced without explanation.

Changed:

Gibbs sampling of the genetic effects α : Although not used in any of the analysis presented here, the sampling of the genetic effects is considered only for completeness sake. After g the genetic effects α are updated. Z is accompanied by either a genomic relationship matrix (GRM) G as specified by VanRaden (44), such as VV' , a $n_R \times n_R$ matrix; or a relationship matrix derived from pedigree relationships. Such as A_{22} , a $n_R \times n_R$ matrix, and which is extracted from the pedigree's A-matrix. For example, let $A = \begin{bmatrix} A_{11} & A_{12} \\ A_{21} & A_{22} \end{bmatrix}$, then the 4 blocks

(sub-matrices) specify the relationships between the individuals A_{11} not in the analysis but related to the individuals in the analysis contained in A_{22} .

Let $a^0 = 0$, $\lambda = \sigma_e^2 / \sigma_a^2$, $\sigma_{a,t}^2 = \sigma_t^2 * h_t^2 * 0.1$ and $\sigma_{g,t}^2 = \sigma_t^2 * h_t^2 * 0.9$. Then in terms of a genomic relationship matrix, G , at each iteration $\forall i: a_i^{l+1} \sim N \left(\frac{(Z'W_e)_i - (G^{-1}\lambda a)_i}{(Z'WZ + G^{-1}\lambda)_{ii}} + a_i^l \frac{\sigma_e^2}{(Z'WZ + G^{-1}\lambda)_{ii}} \right)$ and $e^{l+1} = e^l - Z_i(a_i^{l+1} - a_i^l)$. Here, generically, the subscript $(H)_i$ is the i^{th} element of vector H , while subscript $(H)_{ii}$ implies the i^{th} diagonal element of matrix H .

To:

Gibbs sampling of the genetic effects a : Although not used in any of the analysis presented here, the sampling of the polygenic genetic effects is considered only for completeness and is recommended when the SNPs do not explain all the genetic variance or when it is desired to fit all the SNPs with small effects as well as a small number of SNPs with larger effects. In the former case one can use the A matrix derived from the pedigree and in the latter case a GRM based on all the SNPs (45)

After estimates for g are made the genetic effects a are updated. Z is accompanied by a genomic relationship matrix often extracted from the pedigree known as the A-matrix, both of which are used to account for genetic relationships not accounted for by the SNPs.

Let $a^0 = 0$, $\lambda = \sigma_e^2 / \sigma_a^2$, Then at each iteration l and $\forall i: a_i^{l+1} \sim N \left(\frac{(Z'W_e)_i - (A^{-1}\lambda a)_i}{(Z'WZ + A^{-1}\lambda)_{ii}} + a_i^l \frac{\sigma_e^2}{(Z'WZ + A^{-1}\lambda)_{ii}} \right)$ and $e^{l+1} = e^l - Z_i(a_i^{l+1} - a_i^l)$ and $\lambda = \sigma_e^2 / \sigma_a^2$. Note the notation $(H)_i$ is used to represent the i^{th} element of vector H , while $(H)_{ii}$ represents the i^{th} diagonal element of matrix H . For the starting and initialization, we partition $\sigma_t^2 * h_t^2$ using a 1:9 ratio between $\sigma_{a,t}^2$ and $\sigma_{g,t}^2$ respectively to produce a starting value for $\sigma_{a,t}^2$. Hence, $a^0 = 0$, $\sigma_{a,t}^2 = \sigma_t^2 * h_t^2 * 0.1$ and $\sigma_{g,t}^2 = \sigma_t^2 * h_t^2 * 0.9$.

See lines 696-710

669-676 Here we encounter again the stumbling block mentioned earlier in the review. You claim a multi-trait model but the traits must be assumed uncorrelated!!!! Also, what is a Cholesky trait? The Cholesky transformation is well known, but I do not know what you mean by a "Cholesky trait"???

Cholesky traits are the traits derived from a Cholesky transformation of an initial set of traits. It is shorthand for Cholesky transformed traits.

Changed:

Cholesky traits (25, 46).

To

Cholesky **transformed** traits (25, 47).

See line 736

As explained above a multi-trait analysis can be performed by transforming the original traits to traits that are uncorrelated. We assume that the Cholesky transformation produces traits which are

uncorrelated at the residual level and at the genetic level. This is an approximation but an acceptable one in many cases.

Expressions following 683. Where is all this coming from? It is dropped like a bomb from the sky without any explanation whatsoever.

We have not presented the derivation of these formulae because it has already been published in Kemper KE, Bowman PJ, Hayes BJ, Visscher PM, Goddard ME. A multi-trait Bayesian method for mapping QTL and genomic prediction. *Genet Sel Evol.* 2018;50(1):10

However, we have tried to make it clearer that these are the formulae for sampling J for each SNP. This section now reads:

The multi-trait model for BayesR, was presented by Kemper *et al* (25) but we will describe it briefly for completeness. The model for each trait is the same as described for the single trait case. That is using subscript t to indicate trait $y_t = Xu_t + Vg_t + Za_t + e_t$. It is assumed that the traits are uncorrelated or decorrelated, by being replaced by their respective PCA components, or Cholesky transformed traits (25, 47). However, the multi-trait model includes an indicator variable J . When $J = 0$, the current SNP under consideration has zero effect on all traits. However, if $J = 1$ then the SNP can be associated with one or more traits, and in this case the SNP effect is sampled for each trait independently. Therefore, there is a vector π for each trait which stores the probabilities that $k = 1, 2, 3$ or 4 for that trait. However, the interpretation of π is slightly different to the single trait case because it is the probability conditional on $J = 1$. In BayesR, when a SNP is processed, first J is sampled based on all traits and if $J = 1$ the effect on each trait is sampled independently as in the single trait case.

The multi-trait model requires an additional parameter $\pi^* \sim \text{Beta}(\alpha^*)$, for the probability that $J = 1$. For the current SNP J is sampled $J = 1$ with probability p and $J = 0$ with probability $1 - p$ where (using subscript t for traits and k for the component of the mixture within each trait):

$$p = \frac{1}{1 + e^{(p(\bar{g}|J=0) - p(\bar{g}|J=1))}}$$

$$p(\bar{g}|J = 0) = \log(\pi_0^*) + \sum_t \log(L_{t,0})$$

$$p(\bar{g}|J = 1) = \log(\pi_1^*) + \sum_t \log\left(\sum_k L_{t,k} * \pi_{t,k}\right)$$

$$L_{t,k} = \sqrt{v_{t,k}} * e^{-0.5 * v_{t,k} * \bar{g}_{j,t}^2}$$

$$v_{t,k} = \frac{1}{(\sigma_{k_t}^2 + \sigma_{e_t}^2)}$$

Where \bar{g} is the least square estimate of the effect as specified previously and \bar{g}^2 is its square. $L_{t,k}$ is the likelihood for trait t of SNP j being sampled from distribution k . If $J = 1$, then the sampling of the effect of each trait was as described for the single trait case and each trait was processed independently.

The probabilities that a SNP is in the model (π^*) is sampled at the end of each outer iteration using: $\pi^* \sim \text{Beta}(\alpha^* + \beta^*)$. where $\alpha^* = \{1,1\}$ and $\beta^* = \{\beta_1^*, \beta_2^*\}$ where β_1^* is the number of

SNPs currently not in the model and β_2^* is the number of SNPs currently in the model. The initial values, for the first iteration were set to $\pi^* = \{0.95, 0.05\}$

See line 732-761

688 What is a "proportional probability"?

Changed:

The **proportional** probabilities for a SNP being associated or not is sampled at the end of each outer iteration using: $\pi^* \sim \text{Dirichlet}(\alpha^* + \beta^*)$.

To

The probabilities for a SNP being associated or not is sampled at the end of each outer iteration using: $\pi^* \sim \text{Dirichlet}(\alpha^* + \beta^*)$.

See line 755

691 What is an "uninformed" prior?

This question no longer applies as we have changed:

The proportional probabilities for a SNP being associated or not is sampled at the end of each outer iteration using: $\pi^* \sim \text{Dirichlet}(\alpha^* + \beta^*)$. The initial values, for the first iteration were set to (0.95, 0.05). β^* was used to keep track of the number of SNPs being associated or not during each outer iteration, where $\alpha^* = (1,1)$, is an uninformed Dirichlet prior. Then, a SNP was considered associated if $p_{j,t} \geq x$, where x is a uniform random number drawn from the domain [0,1). For each associated SNP, the probability that a trait t is in a distribution k was sampled from $\frac{L_{t,k} * \pi_{t,k}}{\sum_k L_{t,k} * \pi_{t,k}}$, using the multinomial distribution as given above.

To:

The probabilities that a SNP is in the model (π^*) is sampled at the end of each outer iteration using: $\pi^* \sim \text{Beta}(\alpha^* + \beta^*)$. where $\alpha^* = \{1,1\}$ and $\beta^* = \{\beta_1^*, \beta_2^*\}$ where β_1^* is the number of SNPs currently not in the model and β_2^* is the number of SNPs currently in the model. The initial values, for the first iteration were set to $\pi^* = \{0.95, 0.05\}$

See lines 758-761

694 Where is the multinomial distribution?

The multinomial distributions is the number of SNPs falling into the 4 distribution each iteration we have removed:

the probability that a trait t is in a distribution k was sampled from $\frac{L_{t,k} * \pi_{t,k}}{\sum_k L_{t,k} * \pi_{t,k}}$, using the multinomial distribution as given above

643 Where do 0.1 and 0.9 come from?

This reflects our expectation that maybe 10% of the total genetic variance might be associate with the polygenetic effects as we need some initial estimate of $\sigma_{a,t}^2$ to start the sampling from.

We have added the following line of text:

For the starting and initialization, we partition $\sigma_t^2 * h_t^2$ using a 1:9 ratio between $\sigma_{a,t}^2$ and $\sigma_{g,t}^2$ respectively to produce a starting value for $\sigma_{a,t}^2$. Hence, $a^0 = 0$, $\sigma_{a,t}^2 = \sigma_t^2 * h_t^2 * 0.1$ and $\sigma_{g,t}^2 = \sigma_t^2 * h_t^2 * 0.9$.

See lines 708-710

Response to Reviewer 3:

Dear Reviewer, 3

We would like to thank you for all the time that you obviously spent in the review process on our manuscript. We are most gratefully for all the useful and constructive comments, all of which I believe will go towards strengthening the scientific soundness and improves the communication quality of our work.

With respect to the new manuscript, new text to that of the original document is highlighted in blue while line numbers with such changed text are highlighted with a vertical bar.

Below we give our response, and please note that your questions and comments are in bold font, while our replies are not.

Regards

Edmond Breen.

Reviewer #3 (Remarks to the Author):

Dear authors,

I have finished reading the manuscript and I think it is very interesting.

I have several comments, that can help you to improve the manuscripts, that you can find below.

COMMENTS

Line 45-53, There exists other models that assign mixture distribution to marker effects, for example BayesC, BayesB, which is the advantage of BayesR?.

Given many SNPs, BayesB and BayesC use a prior where some of the SNPs have zero effect.. In Bayes C the remaining SNPs have a normal distribution whereas in Bayes B they have a t-distribution.

BayesR is similar but its prior is based on a mixture of n-Gaussian distributions, where n is usually 4, and where distribution 1 is for the SNPs that have zero effect. This allows BayesR to model a wide range of distributions. For instance, it can model a long-tailed distribution which Bayes C cannot and it is more flexible than Bayes B.

Bayes R was published in 2011 and has been used in several publications since. The purpose of this paper was to present a faster method of computing Bayes R and not to argue that it is better than all alternative models.

Lines 104, 105. How robust is the method when setting different values for the constants that multiply σ_g^2 , i.e., 10^{-4} , 10^{-3} , 10^{-2} ?

We think the results are robust to this choice. These values ultimately influence the number of SNPs falling into each of distributions. If you were to multiple all these values by 10 then you expect less SNPs to be sampled in distributions 2,3 and 4, and hence more SNPs will end up in the NULL distribution and vice a versa if they are multiplied by 10^{-1} .

Line 108, why these hyperparameters? Please explain or add a reference.

We have changed:

The mixing proportions π are assumed to be drawn from a Dirichlet distribution with parameter = (1,1,1,1).

To:

The mixing proportions π are assumed to be drawn from a Dirichlet distribution with parameter = (1,1,1,1), a uniform prior, such that any SNP a priori is equally likely to be assigned to any one of the 4 distributions.

See lines 103-106

Lines 109, 128-130 How to select the number of blocks and the number of markers included in each one?.

We have added:

The number of blocks, n_B , is determined from the block size, n , and is defined as the least integer greater or equal to $\frac{n_M}{n}$.

See lines 110-111

We have also added the supplementary file "Optimizing block size and number of inner cycles.pdf" which shows explicitly how the number of blocks, the block size and the number of inner iterations was determined and specified to the recommendations given in the main manuscript.

See Lines 272-274.

Figure 2, move the legend so that we can see the bars. Perhaps a more informative plot can be built based on estimated π 's for each of the classes. Compare the true π 's with the estimated ones.

Please note Figure 2 is now Figure 3. Also note that the true π are given in the first stacked bar in each plot and is marked as the Exp, for expected.

Figure 1: Stacked bar plots for the mixture components inferred with respect to BayesR configuration (BR, BR3 and BR3+) and iterations (50 to 2000). Y-axis is the log2 of the number of SNPs. The expected (Exp) number of SNPs for each component is given in the first bar in each plot where $Exp = (396000, 3485, 500, 15)$. Top row gives results observed for the 10% heritability data set H10, while the bottom row is for the 30% heritability, H30, data set.

Figure 3, Add an extra figure where you plot the true SNP effect vs Estimated SNP effect.

Note Figure 3 is now Figure 4

While we haven't added an extra figure, we have added to Figure 4 the correlation values for the correlations asked for, and which are seen are above panels b, c and d. Please refer to the Figure legend for explanation.

Figure 2: The true SNP effects for the 4,000 simulated causal variants and their estimated effects using the H30 training data set, embedded within three different genotype densities. All results are from BR3 using five chains, each of length 2,000 and with a block size of 25. The values above **b**), **c**), and **d**), are all Pearson's correlation values. The first value above each of these figures is the correlation between estimated and true breeding values, the 2nd value is the correlation between the true 4,000 causal SNP effect values to their corresponding estimates, while the 3rd value is the estimated SNP effects correlation across all SNPs within each of the analysis to their simulated true values, **b**) = 400,000 SNPs, **c**) = 40,000 SNPs and **d**) = 4,000 SNPs.

Figure 5, Remove the titles from Figure 5 and use the letters a, b to give the description in the legend.

Figure 5 is now Figure 6:

Figure 3 Comparison of the accuracy of genomic prediction and computational efficiency of BayesR3 (BR3) to EM-BayesR (EM) and GBLUP (MTG2 software). Each comparison is for a single trait analysis for milk, fat, and protein yield, using a reference set of 25,000 Holstein and Jersey cattle. Accuracy was tested in 3 validation sets: 398 Jersey bulls, 702 Holstein bulls and 3,082 RDC cows and 212 RDC Bulls. **a**) accuracy of genomic prediction as a function of trait and breed. Computation requirements in terms of **b**) runtime and **c**) memory requirements

Line 564, "Z is an identity matrix", well if the phenotypical records and the rows/columns of matrix A are properly sorted, yes, but is not always the case. Also in other contexts (e.g. plant breeding) you can have replicates, so Z not necessarily is an identity matrix.

We Agree. But in our examples Z=I. We do not wish the complicate the presentation any further by allowing for a general Z.

Lines 569-570, replace "*" with the times operator so that you have the same notation in all the document.

Done. The equation now reads as:

$$\sigma_k^2 = \begin{cases} 0 \times \sigma_g^2 & \text{with probability } \pi_1 \\ 0.0001 \times \sigma_g^2 & \text{with probability } \pi_2 \\ 0.001 \times \sigma_g^2 & \text{with probability } \pi_3 \\ 0.01 \times \sigma_g^2 & \text{with probability } \pi_4 \end{cases}$$

See Line 622

Lines 570-571: Describe σ_t^2 . Which is the prior assigned to σ_g^2 ?

The following text has been added:

The trait variance σ_t^2 is the estimated trait variance obtained from the data, and its exact form is given below (see Material and Methods: Weighted Analysis).

See lines 630-631

Lines 572-573, the use on improper priors can cause convergency issues, comment about this, and give reference to justify the selection of hyperparameters. Furthermore, there are two parameterizations for scaled inverse chi-squared distributions (e.g. Sorensen & Gianola, 2002). Which is the one used here?

Changed:

The priors for σ_a^2 and σ_e^2 are scaled inverse chi-squares with scale 0 and degrees of freedom -2

To:

The priors for σ_a^2 and σ_e^2 are scaled inverse chi-squares with uniform priors (44).

Line 629

Matrix W has not been described... in the weighted analysis you describe a matrix of weights, I assume that in the case of univariate analysis W is the identity matrix

W is a matrix with weights for each record on the diagonal and zero elsewhere. Therefore, each trait has its own W matrix.

see lines 638-645

References:

Sorensen, Daniel, and Daniel Gianola. 2002. Likelihood, Bayesian and MCMC Methods in Quantitative Genetics. Statistics for Biology and Health. New York: Springer-Verlag.

added to line 629

The priors for σ_a^2 and σ_e^2 are scaled inverse chi-squares with uniform priors (44).

Added Reference:

44. Sorensen D, Gianola D, Gianola D. Likelihood, Bayesian and MCMC Methods in Quantitative Genetics. New York: Springer-Verlag; 2002.

Line 963

Standardize the notation in mathematical expressions, sometimes the matrixes/vectors appear with boldface font, and in other cases without boldface, for example equations in lines 569-573.

All bold face font with respect to maths notation has been removed. See equivalent section lines 619-625

How do you check convergence to the stationary distributions for BayesR3?

We have added a new section called "Convergence" line 167

We advocate that convergence can be checked by comparing the correlation of the SNP effects across chains, and for example, for 5 chains, the mean of 10 correlations is inspected, as shown in the new **Error! Reference source not found.**

Figure 4. Across chains convergence analysis. Mean across chain Pearson's correlations for iteration {50,100,200,500,1000,2000 and 5000} for the analysis and data presented in **Error! Reference source not found.b.** The number of chains was 5, therefore each plotted point represents the mean of 10 correlations.

See Lines 167-188.

Can you add a section with 5 cross-validation analysis to show how well BayesR3 predicts?. You can compare it with GBLUP for example.

In dairy cattle, randomised k-fold cross-fold validation is generally not appropriate for several reasons. For example, the training set includes both bulls (progeny test milk phenotypes) and cows (their own milk phenotypes). Thus, it is important that offspring and parent pairs are not split across the training and validation sets to ensure independent tests.

Therefore, we opted a more stringent validation approach than cross-fold validation. We have used three independent validation sets from a younger cohort of individuals, with no offspring-parent pairs split across the training and validation. Each validation set comprised a different cattle breed (like different ethnic populations in humans). While two of the breeds were included in the training population, the 3rd breed was not. Thus, we validated in diverse populations from closer to more genetically distant. Additionally, three traits were tested for prediction accuracy in each population, and the speed and memory requirements of the analyses were also compared.

Line 736: Code availability

The code is not available, what is available is the software.

Nowadays it is a widely used practice to provide the source code that implements proposed models, so that others can adapt the programs for other problems. Are you planning to upload the code and make it available to the public using one of the existing licenses (e.g. GNU-GPL, BSD, etc.)?. In the other hand, if you just provide the executables, it will be difficult to run that executable because very often linux distributions do not include the same library versions and are not always installed in the same directories, so one executable that was generated using Linux Centos, will not run on Linux Ubuntu, etc.

My suggestion is that you upload the code to a public site, e.g. github.

We are now supplying R code for doing a single trait blocked Bayes R3 analysis, see supplementary file BayesR3.zip. This explicitly shows how to determine and set up the blocks and how to do our style of blocked Gibbs sampling. It also gives all the needed matrix operations compatible with the C++ Eigen algebraic library and the probability mathematics translated into code that helps to minimize the possibilities of numeric over and underflow.

We also offer the binary executable to anyone who asks for it, and it will run on any intel-based processor regardless of Linux version of runtime libraries. Please note that it is not correct in saying that our binary executable will only run on the version of Linux we build it on. All that needs to be done, and as we have done, is to build a statically linked binary.

REVIEWERS' COMMENTS:

Reviewer #1 (Remarks to the Author):

The authors have answered all my questions. When I read the manuscript again, I found one potential mistake. For line 685 where you show how errors are updated, do you need W in the equation? Should it be $e < -e - V_b'(g_b^{\{l+1\}} - g_b^{\{l\}})$? I have no other comments.

Reviewer #3 (Remarks to the Author):

Dear authors,

I have reviewed your response letter, there are some issues there for example in the pdf of the file some references were not found, "Error! Reference not found."

I think that you need to better justify the selection of constants that multiply σ^2_g , i.e., 10^{-4} , 10^{-3} , 10^{-2} , etc. I understand your response, but is there a more robust alternative?, why don't you assign a prior distribution to the different variance parameters in each component?, please mention this issue at least in the discussion section.

Lines 93, 613, remove Z from your model, in both cases Z is the identity matrix. If your software is able to deal with replicates then you can mention that in a more general setting your model and your software is able to deal with this case.

Regards.

Response to Reviewer 1:

Dear Reviewer, 1

We would like to thank you again for your 2nd review of our manuscript. We are most gratefully for your useful correction for one of our equations.

We have now revised the manuscript according to Communications Biology specifications. We moved Tables 3, 4 and 5 to the supplementary information. They are referenced in the manuscript as Supplementary Table 1, 2 and 3 respectively. Therefore, we have added a Supplementary information file which contains the above-mentioned tables plus:

Supplementary Note 1 contains a more detailed description of the Mixture Model Distribution and its allocation values.

Supplementary Note 2 contains the previous information from Supplementary file: determining the optimal block size and number of inner iterations.

We renamed our BayesR3.zip file to Supplementary Software 1.zip

We renamed our File1.xlsx to Supplementary Data 1.xlsx

Our Statistical Analysis section has been renamed to Statistics and Reproducibility

We have changed the following text:

Where σ_g^2 and is the additive genetic variance explained by the SNPs cumulatively. The mixing proportions π are assumed to be drawn from a Dirichlet distribution with parameter = (1,1,1,1), a uniform prior, such that any SNP a priori is equally likely to be assigned to any one of the 4 distributions.

To read:

Where σ_g^2 is the additive genetic variance explained by the SNPs cumulatively and is estimated from the data. The choice of 4 distributions, is historical (13), but any number of distributions can be included in the mixture if needed. For example, it has been reported that adding the variance group $10^{-5} \times \sigma_g^2$ can help identify SNPs with very small effects if the dataset is very large (26). Therefore, the allocations values (0, 10^{-4} , 10^{-3} , 10^{-2}) seen in Equation (2) are held constant and used to scale the genetic variance and to help fit long tailed distributions as discussed in Supplementary section: Mixture Model Distributions. The 10x scaling between the allocation values allows the distributions generated to be relatively smooth and effects can shuffle from one distribution to the next between MCMC cycles. The mixing proportions π are also estimated from the data and are assumed to be drawn from a Dirichlet distribution with parameter = (1,1,1,1), a uniform prior, such that any SNP a priori is equally likely to be assigned to any one of the 4 distributions.

see lines 101 to 112

We have changed:

Where σ_g^2 and is the additive genetic variance explained by the SNPs cumulatively. The mixing proportions π are assumed to be drawn from a Dirichlet distribution with parameter = (1,1,1,1), a uniform prior, such that any SNP a priori is equally likely to be assigned to any one of the 4 distributions.

To read:

Where σ_g^2 is the additive genetic variance explained by the SNPs cumulatively and is estimated from the data. The choice of 4 distributions, is historical (13), but any number of distributions can be included in the mixture if needed. For example, it has been reported that adding the variance group $10^{-5} \times \sigma_g^2$ can help identify SNPs with very small effects if the dataset is very large (26). Therefore, the allocations values (0, 10^{-4} , 10^{-3} , 10^{-2}) seen in Equation (2) are held constant and used to scale the genetic variance and to help fit long tailed distributions as discussed in Supplementary section: Mixture Model Distributions. The 10x scaling between the allocation values allows the distributions generated to be relatively smooth and effects can shuffle from one distribution to the next between MCMC cycles. The mixing proportions π are also estimated from the data and are assumed to be drawn from a Dirichlet distribution with parameter = (1,1,1,1), a uniform prior, such that any SNP a priori is equally likely to be assigned to any one of the 4 distributions.

see lines 101 to 112

Below is our feedback to your review

Please note your comments and questions are given below in bold.

Reviewer #1 (Remarks to the Author):

For line 685 where you show how errors are updated, do you need W in the equation? Should it be $e \leftarrow e - V_b'(g_b^{l+1} - g_b^l)$? I have no other comments.

Thank you for highlighting this error in our equation

$$e \leftarrow e - V_b'W(g_b^{l+1} - g_b^l)$$

It now reads as:

$$e \leftarrow e - V_b'(g_b^{l+1} - g_b^l)$$

See Line 680.

Best Regards

Edmond J. Breen

Response to Reviewer 3:

Dear Reviewer, 3

We would like to thank you again for your 2nd review of our manuscript. We are gratefully for your useful comments and suggestions.

We have now revised the manuscript according to Communications Biology specifications. We moved Tables 3, 4 and 5 to the supplementary information. They are referenced in the manuscript as Supplementary Table 1, 2 and 3 respectively. Therefore, we have added a Supplementary information file which contains the above-mentioned tables plus:

Supplementary Note 1 contains a more detailed description of the Mixture Model Distribution and its allocation values.

Supplementary Note 2 contains the previous information from Supplementary file: determining the optimal block size and number of inner iterations.

We renamed our BayesR3.zip file to Supplementary Software 1.zip

We renamed our File1.xlsx to Supplementary Data 1.xlsx

Our Statistical Analysis section has been renamed to Statistics and Reproducibility

We have corrected an error in the following equation:

$$e \leftarrow e - V_b' W (g_b^{l+1} - g_b^l)$$

It now reads as:

$$e \leftarrow e - V_b' (g_b^{l+1} - g_b^l)$$

See Line 614.

Below is our feedback to your review.

Please note your comments and questions are given below in bold.

Reviewer #1 (Remarks to the Author):

I have reviewed your response letter, there are some issues there for example in the pdf of the file some references were not found, "Error! Reference not found."

We apologies for this, and any inconvenience it may have caused you. This was a problem that went, unfortunately, unnoticed. It is related to coping and pasting figures between documents, and then Word auto updating the figure references.

I think that you need to better justify the selection of constants that multiply σ^2_g , i.e., 10^{-4} , 10^{-3} , 10^{-2} , etc. I understand your response, but is there a more robust alternative?,

why don't you assign a prior distribution to the different variance parameters in each component?, please mention this issue at least in the discussion section.

We have changed:

Where σ_g^2 and is the additive genetic variance explained by the SNPs cumulatively. The mixing proportions π are assumed to be drawn from a Dirichlet distribution with parameter = (1,1,1,1), a uniform prior, such that any SNP a priori is equally likely to be assigned to any one of the 4 distributions.

To read:

Where σ_g^2 is the additive genetic variance explained by the SNPs cumulatively and is estimated from the data. The choice of 4 distributions, is historical (13), but any number of distributions can be included in the mixture if needed. For example, it has been reported that adding the variance group $10^{-5} \times \sigma_g^2$ can help identify SNPs with very small effects if the dataset is very large (26). Therefore, the allocations values (0, 10^{-4} , 10^{-3} , 10^{-2}) seen in Equation (2) are held constant and used to scale the genetic variance and to help fit long tailed distributions as discussed in Supplementary section: Mixture Model Distributions. The 10x scaling between the allocation values allows the distributions generated to be relatively smooth and effects can shuffle from one distribution to the next between MCMC cycles. The mixing proportions π are also estimated from the data and are assumed to be drawn from a Dirichlet distribution with parameter = (1,1,1,1), a uniform prior, such that any SNP a priori is equally likely to be assigned to any one of the 4 distributions.

see lines 101 to 112

We have also added Note 1 to the supplementary information that contains a more detailed description of the Mixture Model Distribution and its allocation values.

Lines 93, 613, remove Z from your model, in both cases Z is the identity matrix. If your software is able to deal with replicates, then you can mention that in a more general setting your model and your software is able to deal with this case.

We do point out that $Z=I$ in our examples. However, we prefer to keep the Z in the equation for two reasons – It makes the model more general regardless of whether $Z=I$ in our examples and it clarifies for the reader that aspect of the equations. The Z in the equations makes it clear that this term is derived from the polygenic term in the model. Also, the R code supplied with the manuscript comes with a data set containing in part 10000 SNPs, 2326 phenotypes and a pedigree for 3226 simulated animals.

Best Regards

Edmond J. Breen